# Dual expression of Atoh1 and Ikzf2 promotes transformation of adult cochlear supporting cells into outer hair cells

Suhong Sun[1,2†], Shuting Li[1,2†], Zhengnan Luo[1,2†], Minhui Ren[1,2], Shunji He[1], Guangqin Wang[1,2], Zhiyong Liu[1,3]*

[1]Institute of Neuroscience, State Key Laboratory of Neuroscience, CAS Center for Excellence in Brain Science and Intelligence Technology, Chinese Academy of Sciences, Shanghai, China; [2]University of Chinese Academy of Sciences, Beijing, China; [3]Shanghai Center for Brain Science and Brain-Inspired Intelligence Technology, Shanghai, China

**Abstract** Mammalian cochlear outer hair cells (OHCs) are essential for hearing. Severe hearing impairment follows OHC degeneration. Previous attempts at regenerating new OHCs from cochlear supporting cells (SCs) have been unsuccessful, notably lacking expression of the key OHC motor protein, Prestin. Thus, regeneration of Prestin+ OHCs represents a barrier to restore auditory function in vivo. Here, we reported the successful in vivo conversion of adult mouse cochlear SCs into Prestin+ OHC -like cells through the concurrent induction of two key transcriptional factors known to be necessary for OHC development: *Atoh1* and *Ikzf2*. Single-cell RNA sequencing revealed the upregulation of 729 OHC genes and downregulation of 331 SC genes in OHC-like cells. The resulting differentiation status of these OHC-like cells was much more advanced than previously achieved. This study thus established an efficient approach to induce the regeneration of Prestin+ OHCs, paving the way for in vivo cochlear repair via SC transdifferentiation.

*For correspondence:
Zhiyongliu@ion.ac.cn

†These authors contributed equally to this work

## Introduction

Hair cells (HCs) are the auditory sensors that enable vertebrates to hear. These cells are located in the auditory epithelium, the latter referred to as the organ of Corti (OC) (*Wu and Kelley, 2012*). Located near HCs are several supporting cell (SC) subtypes, which, from the medial side to lateral side, are Pillar cells (PCs) and Deiters' cells (DCs) (*Montcouquiol and Kelley, 2020*). Auditory HCs comprise two subtypes, inner and outer HCs (IHCs and OHCs). IHCs are the primary sensory cells and innervated by type-I cochlear spiral ganglion neurons (SGNs). IHCs specifically express vGlut3, encoded by *Slc17a8*, which is required for sound information transmission from IHCs to SGNs (*Li et al., 2018b*); consequently, *Slc17a8−/−* mice are completely deaf (*Ruel et al., 2008*; *Seal et al., 2008*). In contrast, OHCs act as sound amplifiers and lack vGlut3. OHCs express a motor protein unique to them, Prestin, encoded by *Slc26a5* (*Zheng et al., 2000*). Prestin mediates electromotility, which enables OHCs to function as sound amplifiers, an important step in broadening our ability to perceive sounds. Indeed, Prestin-deficient (*Slc26a5−/−*) mice show severe hearing impairment (*Liberman et al., 2002*). IHCs and OHCs are believed to originate from communal pool of Atoh1+ progenitor cells (*Driver et al., 2013*; *Groves et al., 2013*; *Matei et al., 2005*; *Tateya et al., 2019*).

Atoh1 is a bHLH transcriptional factor (TF) that is necessary for specifying general HC fate, and, accordingly, both IHCs and OHCs are lost in *Atoh1−/−* mice (*Bermingham et al., 1999*). Two additional

TFs, encoded by Insm1 and Ikzf2, are known to be required for specifying OHC fate and/or repressing IHC fate (*Chessum et al., 2018*; *Wiwatpanit et al., 2018*). Furthermore, OHCs in $Insm1^{-/-}$ and Ikzf2 point-mutant mice tend to transdifferentiate into IHCs. Insm1 and Ikzf2 differ in that the former is only transiently expressed in differentiating OHCs (*Lorenzen et al., 2015*), whereas the latter is expressed permanently throughout the lifetime of OHCs (*Chessum et al., 2018*). Unlike IHCs, OHCs are highly vulnerable to ototoxic drugs, noise, and aging and their damage is a common cause of hearing impairment. Non-mammalian vertebrates such as fish and chicken are able to regenerate HCs from neighboring SCs, during which key HC developmental genes (e.g. *Atoh1*) are reactivated (*Atkinson et al., 2015*). Mammals have lost this regenerative capacity (*Janesick and Heller, 2019*). Nonetheless, recent single-cell transcriptomic analyses of cochlear cells suggest that some SCs share recent progenitors with OHCs (particularly Lgr5+ PCs and DCs populations) and are thus regarded as a promising source for new OHCs (*Chai et al., 2012*; *Kolla et al., 2020*; *McLean et al., 2017*). Importantly, PCs and DCs are also in close physical proximity to OHCs requiring minimal if any displacement.

In previous work, we demonstrated that in vivo ectopic *Atoh1* expression in neonatal and juvenile SCs (primarily PCs and DCs) is sufficient to convert these cells into nascent HCs that express early HC markers such as Myo6 and Myo7a (*Liu et al., 2012a*). In contrast, adult PCs and DCs are not sensitive to ectopic Atoh1 expression (*Kelly et al., 2012*; *Liu et al., 2012a*), unless additional manipulations are performed (*Walters et al., 2017*). Despite this progress, no newly in vivo transformed HCs has been reported to express Prestin (*Chai et al., 2012*; *Liu et al., 2012a*; *Walters et al., 2017*). Thus, an outstanding challenge is how to regenerate Prestin+ OHCs from SCs, particularly from adult lesioned cochlea; indeed, OHC damage is the much more prevalent in the elderly population, than in younger generations. As ectopic Ikzf2 induction in IHCs leads to ectopic Prestin expression (*Chessum et al., 2018*), we hypothesized that together Atoh1 and Ikzf2 might synergistically reprogram adult PCs and DCs into not only HCs, but into further differentiated Prestin+ OHCs.

To test this hypothesis, we generated genetic mice lines that allowed us to conditionally and concurrently induce ectopic expression of Atoh1 and Ikzf2 in adult cochlear SCs (primarily PCs and DCs), with or without pre-damaging wild-type OHCs. In summary, the results that emanated from our comprehensive genetic, transcriptomic, immunostaining, morphological, and fate-mapping analyses supported our general hypothesis. Indeed, new Prestin+ OHC -like cells were frequently observed across the entire cochlear duct; interestingly, this occurred much more readily when endogenous OHCs were pre-damaged. To the best of our knowledge, this is the first report of in vivo generation of Prestin+ OHC -like cells from adult cochlear SCs. Our findings identify Atoh1 and Ikzf2 as potential targets for OHC regeneration therapy in hearing-impaired patients. Nonetheless, future work is still called for as the Prestin+ OHC -like cells were still insufficiently differentiated to restore hearing loss caused by OHC damage.

## Results

### Generation of conditional Ikzf2-overexpression mouse model

To test whether concurrent Atoh1 and Ikzf2 expression was sufficient to induce transdifferentiation of adult SCs into OHCs, we needed various conditional transgenic lines. A conditional Atoh1-overexpression transgenic strain, CAG-Loxp-stop-Loxp-*Atoh1*\*2xHA+ (CAG-LSL-*Atoh1*+ for short) already existed, and its ability to efficiently drive high Atoh1 protein expression was previously confirmed (*Liu et al., 2012a*). In this mouse line, 2× HA fragments were tagged to the C-terminus of Atoh1 as a practical means for reading out expression. However, a similar conditional expression model for Ikzf2 was not available. Therefore, we began by generating a *Rosa26*-CAG-Loxp-stop-Loxp-*Ikzf2*\*3xHA-T2A-Tdtomato/+ knock in mouse line ($Rosa26^{CAG-LSL-Ikzf2/+}$ for short), in which Ikzf2 is similarly tagged with 3× HA fragments at its C-terminus (*Figure 1—figure supplement 1A–C*). The mouse line was generated following the CRISPR/Cas9 gene-targeting approach (*Li et al., 2020b*), Southern blotting results confirmed the lack of random donor DNA insertion into the host mouse genome (*Figure 1—figure supplement 1D,E*), and PCR-genotyping of tail DNA allowed us to readily distinguish among wild-type, heterozygous, and homozygous littermates (*Figure 1—figure supplement 1F*). Both heterozygous and homozygous $Rosa26^{CAG-LSL-Ikzf2/+}$ mice were healthy and fertile and did not display any noticeable abnormalities.

The genetic constructs were designed such that upon Cre-mediated recombination, Tdtomato and Ikzf2 would be translated, initially together from the same polycistronic mRNA, and subsequently cleaved by the 2A peptide (*Li et al., 2020a*); and once triggered, expression of Ikzf2 and Tdtomato would be permanent, enabling genetic fate-mapping analysis at single-cell resolution. Thus, this construct, when used in conjunction with a SC-specific Cre-driver, would readily allow us to distinguish potential new OHCs (Tdtomato+) derived from adult cochlear SCs (primarily PCs and DCs in this study) from endogenous OHCs (Tdtomato−). Due to the lack of high-quality commercial antibodies against either Atoh1 or Ikzf2, we used anti-HA antibody to assess the presence of ectopic Atoh1 and Ikzf2 proteins in CAG-LSL-*Atoh1*+ and *Rosa26*$^{CAG-LSL-Ikzf2/+}$ strains, respectively.

## Induction of ectopic Ikzf2 in IHCs is sufficient to induce OHC-markers, prestin, and Ocm expression, as well as reduce IHC-marker, Slc7a14 expression

We began by validating the ability of the new mouse model (*Rosa26*$^{CAG-LSL-Ikzf2/+}$) to drive ectopic and 'functional' Ikzf2 expression. It is known that IHCs, infected with an Anc80-virus expressing Ikzf2 (*Chessum et al., 2018*), begin synthetizing the typically OHC-exclusive protein, Prestin (*Fang et al., 2012*; *Liberman et al., 2002*; *Zheng et al., 2000*). We therefore used ectopic Prestin expression as a proxy for 'functional' Ikzf2 expression in IHCs. More specifically, we crossed the transgenic Atoh1-CreER+ mouse line, in which Cre activity is restricted to HCs (both IHCs and OHCs) (*Chow et al., 2006*; *Cox et al., 2012*), to the new *Rosa26*$^{CAG-LSL-Ikzf2/+}$ line. Atoh1-CreER+ or *Rosa26*$^{CAG-LSL-Ikzf2/+}$ mice were used as controls, and Atoh1-CreER+; *Rosa26*$^{CAG-LSL-Ikzf2/+}$ mice were experimental group mice. All groups were administered tamoxifen at postnatal day 0 (P0) and P1, and sacrificed and analyzed at P42 (*Figure 1A*, *Figure 1—figure supplements 2 and 3*).

We focused our analysis on the IHC Prestin, HA (Ikzf2), and Tdtomato expression patterns, but we did note that OHCs were also directly affected by the manipulation. In the Atoh1-CreER+ control group (n = 3), neither Prestin nor Tdtomato was expressed in IHCs at P42 (*Figure 1B–B'''*). In contrast, in the experimental P42 mice (n = 3) Prestin was ectopically expressed in IHCs that were vGlut3+/Tdtomato+ (arrows in *Figure 1C–C'''*) and, as expected, not in IHCs that were vGlut3+/Tdtomato− (asterisks in *Figure 1C–C'''*). All Tdtomato+ IHCs were Prestin+ and all Prestin+ IHCs were Tdtomato+. Quantification revealed that 30.8% ± 4.2%, 38.4% ± 1.6%, and 70.7% ± 6.0% of IHCs were Prestin+ in basal, middle, and apical turns, respectively, at P42 (*Figure 1D*, *Figure 1—figure supplement 2*); the apical turn thus harboring more Prestin+ IHCs than the basal and middle turns. This is in accord with the higher reported Cre efficiency of Atoh1-CreER+ in the apex of the cochlea (*Cox et al., 2012*). We also confirmed that all Tdtomato+ (Prestin+) IHCs expressed HA (Ikzf2) and vice versa (*Figure 1E,F*), which validated the co-expression of Ikzf2 and Tdtomato. As a point of potential interest, these results bring further support toward the ability of Ikzf2 to induce Prestin expression in IHCs. We noted that despite Tdtomato+ OHCs likely expressing both endogenous and ectopic Ikzf2, they appeared normal suggesting that these cells are able to tolerate additional Ikzf2 expression, at least until P42 (*Figure 1C–C''' and F*).

Besides Prestin, overexpressing Ikzf2 IHCs also upregulated a second OHC-specific marker, Oncomodulin (Ocm) (*Simmons et al., 2010*; *Tong et al., 2016*). This was only observed in the experimental mice (n = 3, P42), but not in the control *Rosa26*$^{CAG-LSL-Ikzf2/+}$ mice (n = 3, P42) (*Figure 1—figure supplement 3A-B''*). Again, all Tdtomato+ IHCs were Ocm+ and vice versa. Moreover, triple staining against the IHC markers, Slc7a14 (*Li et al., 2018a*), Ctbp2 (a marker of synaptic ribbons) (*Buran et al., 2010*; *Khimich et al., 2005*; *Liu et al., 2014a*), showed that in the control *Rosa26*$^{CAG-LSL-Ikzf2/+}$ group (n = 3, P42) Slc7a14 was expressed in IHCs and that the cells also contained Ctbp2+ puncta (*Figure 1—figure supplement 3C–C'''*). In contrast, in the experimental mice (n = 3, P42) Slc7a14 expression was diminished and the number of Ctbp2+ puncta was reduced in Tdtomato + IHCs (arrows in *Figure 1—figure supplement 3D–D''*). In Tdtomato− IHCs, Slc7a14 level and number of Ctbp2 puncta were comparable to those of the control mice as expected (dotted line in *Figure 1—figure supplement 3D–D'''*). Lastly, we noted that the size of the cell nuclei, which can be estimated by the area covered by Ctbp2, was smaller in Tdtomato+ IHCs than that in Tdtomato− IHCs.

Taken together, these results show that the *Rosa26*$^{CAG-LSL-Ikzf2/+}$ strain was able to drive sufficient ectopic Ikzf2 expression to induce endogenous IHCs to gain and lose some OHC and IHC characteristics respectively; more specifically, the OHC markers, Prestin and Ocm, were found to be

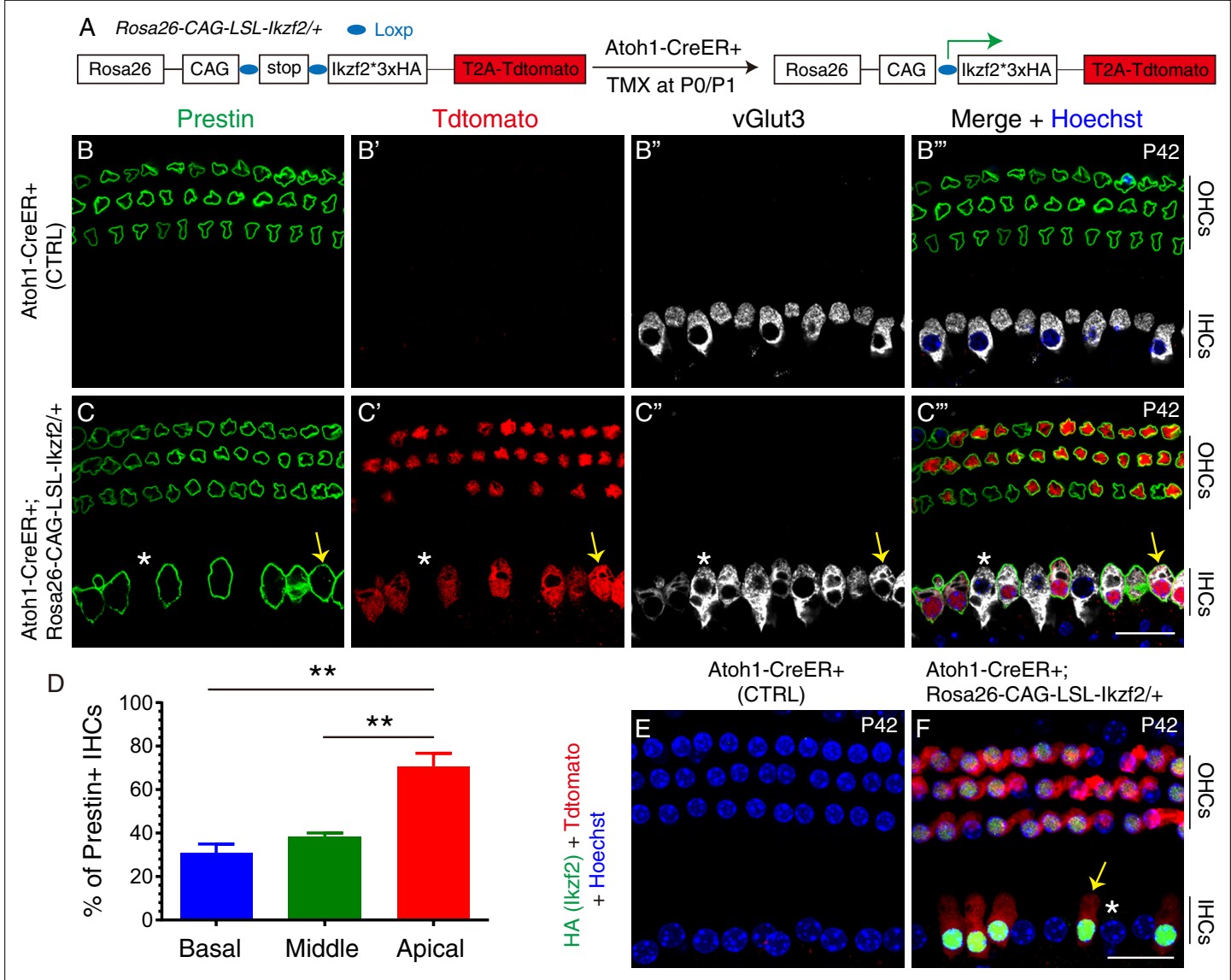

**Figure 1.** Prestin was expressed in IHCs upon ectopic Ikzf2 induction. (**A**) Illustration of approach used to induce Ikzf2 expression in HCs (both IHCs and OHCs) using the Cre-LoxP system. The Atoh1-CreER+ line was used to efficiently drive Cre expression in HCs at neonatal ages when Tamoxifen (TMX) was given at P0 and P1. Tdtomato and Ikzf2 (tagged with HA) were placed in close proximity in the construct to ensure joint expression. (**B–C‴**) Triple labeling against Prestin, Tdtomato, and vGlut3 in P42 cochlear samples: control Atoh1-CreER+ (**B–B‴**) and experimental Atoh1-CreER+; *Rosa26$^{CAG-LSL-Ikzf2/+}$* (**C–C‴**). In wild-type (**B–B‴**) Prestin was only expressed in OHCs; in the experimental group (**C–C‴**), expression was observed in both OHCs and some IHCs. Arrows in (**C–C‴**): Tdtomato+/vGlut3+/Prestin+ IHC; asterisks in (**C–C‴**): vGlut3+/Tdtomato-/Prestin- IHC. (**D**) Quantification of Prestin+ IHCs. More Prestin+ IHCs were present in the apical turn than in basal and middle turns of the cochleae. Data was presented as mean ± SEM, **p<0.01 (Student's t test). (**E, F**) Co-staining against HA (Ikzf2) and Tdtomato in control (**E**) and experimental (**F**) mice at P42. HA+/Tdtomato + cells were present in experimental mice only. Arrow/asterisk in (**F**): IHC with/without HA (Ikzf2) expression. All Tdtomato+ cells were HA+ (Ikzf2 expressing) cells, and vice versa. Scale bar: 20 μm.

The online version of this article includes the following figure supplement(s) for figure 1:

**Source data 1.** It contained the numerical values of the data plotted in the graph of *Figure 1D*.

**Figure supplement 1.** Generation of *Rosa26$^{CAG-LSL-Ikzf2/+}$* knock in mouse line using CRISPR/Cas9 technology.

**Figure supplement 2.** Prestin was expressed in IHCs upon ectopic Ikzf2 induction.

**Figure supplement 3.** IHCs with ectopic Ikzf2 expression activated Ocm and suppressed Slc7a14 expression, as well as exhibited fewer ribbon synapses.

over-expressed, and IHC marker, Slc7a14, downregulated in the Tdtomato+/overexpressing Ikzf2 IHCs. In summary, we confirmed that *Rosa26^CAG-LSL-Ikzf2/+* strain could be used as a powerful genetic model for inducing functional Ikzf2 expression in Cre-expressing cells.

## Induction of *Ikzf2* alone fails to convert adult PCs and DCs into HCs

We next determined whether Ikzf2 alone could reprogram adult cochlear SCs into Prestin+ OHCs. Fgfr3-iCreER+; *Rosa26*-Loxp-stop-Loxp-Tdtomato (Ai9)/+ (Fgfr3-Ai9 for short) mice were administered tamoxifen at P30/P31 and sacrificed at P60 (arrows in *Figure 2A–A'''*). As expected, no Tdtomato+ HCs (neither IHCs nor OHCs) were observed, whilst most SCs (primarily PCs and DCs) were Tdtomato+ (inset in *Figure 2A'*); this was congruent with our previous reports that Fgfr3-iCreER is SC specific at adult ages (*Liu et al., 2012a*; *Liu et al., 2012b*). We also confirmed that induction of Atoh1 alone failed to convert adult cochlear SCs into HCs by analyzing Fgfr3-iCreER+; CAG-LSL-*Atoh1*+ mice (Fgfr3-Atoh1 for short) that were also administered tamoxifen at P30/P31 and sacrificed at P60 (*Figure 2B–B'''*). Despite high levels of ectopic Atoh1 (tagged with HA) were observed in adult SCs (inset in *Figure 2B*), SCs remained Prestin- (*Figure 2B''*). More extensive information regarding *Atoh1* expression patterns in CAG-LSL-*Atoh1*+ mice is available in our previous study (*Liu et al., 2012a*).

We next tested whether driving *Ikzf2* alone in SCs was sufficient to induce trans-differentiation into OHCs. To assess for sufficiency, Fgfr3-iCreER+; *Rosa26^CAG-LSL-Ikzf2/+* mice (Fgfr3-Ikzf2 for short) were administered tamoxifen at P30/P31 and analyzed at P60. Both the HC layer (*Figure 2C–C'''*) and the SC layer (*Figure 2D–D'''*) were imaged. As expected, numerous Tdtomato+ cells were observed in cochleae. As before, all Tdtomato+ cells expressed HA and vice versa (arrows in *Figure 2C–D'''*). However, none of these Tdtomato+/HA+ cells expressed Prestin (arrows in *Figure 2C–D'''*). Furthermore, we could not detect any Tdtomato+/Myo6 (HC marker)+ cells. We also noted the loss of endogenous OHCs (Prestin+/Tdtomato−) throughout the cochlear duct, particularly at the basal turn, which was likely a secondary effect of ectopic Ikzf2 expression in adult cochlear SCs. Collectively, these results suggested that Ikzf2 alone was insufficient for converting adult cochlear SCs into nascent Myo6+ HCs or Prestin+ OHCs. Thus, in terms of cell-fate conversion, more barriers from adult cochlear SCs into OHCs needed to be overcome than from IHCs to OHCs.

## Concurrent induction of Ikzf2 and Atoh1 converts, at low efficiency, adult PCs, and DCs into Prestin+ OHC-like cells

Considering the synergistic effects reported among TFs, such as between Six1, Atoh1, Pou4f3, and Gata3 and between Myc and Notch signaling (*Chen et al., 2021*; *Costa et al., 2015*; *Menendez et al., 2020*; *Shu et al., 2019*; *Walters et al., 2017*), we hypothesized that concurrent induction of Ikzf2 and Atoh1 might convert adult cochlear SCs into OHCs. We tested this by analyzing cochlear samples from Fgfr3-iCreER+; CAG-LSL-*Atoh1*+; *Rosa26^CAG-LSL-Ikzf2/+* mice (Fgfr3-Atoh1-Ikzf2 for short) that were given tamoxifen at P30/P31 and sacrificed at P60. As before, Tdtomato+ SCs were abundant within the OC, and 88.0% ± 2.7%, 94.1% ± 4.3%, and 98.2% ± 1.8% of HA+ cells were Tdtomato+ in the basal, middle, and apical turns, respectively (n = 3). The fact that most Tdtomato+ cells were HA+ further confirmed the high Cre activity in the Fgfr3-iCreER model. The small fraction of HA+/Tdtomato− cells represented populations in which ectopic Atoh1, but not Ikzf2 was expressed (HA is tagged to both Atoh1 and Ikzf2 constructs) due to independent Cre-recombination events in the two loci. No Tdtomato+/HA− cells were observed as Tdtomato and HA are closely appositioned in the *Rosa26^CAG-LSL-Ikzf2/+* model.

Unlike in the case of exclusive Ikzf2 induction (*Figure 2C–D'''*), in Fgfr3-Atoh1-Ikzf2 mice (n = 3), Tdtomato+/Prestin+ cells were occasionally observable (arrows in *Figure 2E–F'''*) at P60. These cells were defined as new OHC-like cells as they were derived from the original Tdtomato+ cochlear SCs (PCs and DCs) but were not identical to wild-type adult OHCs (see below). These OHC-like cells were distributed in both HC (*Figure 2E–E'''*) and SC layers (*Figure 2F–F'''*). However, the number of new OHC-like cells was low, averaging to 4.7 ± 4.2, 0.7 ± 0.3, and 14.3 ± 6.4 in the basal, middle, and apical turns, respectively (*Figure 2G*). Moreover, Prestin protein expression was substantially lower in these OHC-like cells (blue arrows in *Figure 2E–F'''*) compared to wild-type endogenous OHCs, the latter which expressed Prestin but not Tdtomato (yellow arrows in *Figure 2F–F'''*). As observed before, endogenous OHC loss occurred throughout the cochlear ducts (*Figure 2E–F'''*). These results supported the conclusion that concurrent induction of Atoh1 and Ikzf2 was sufficient to reprogram,

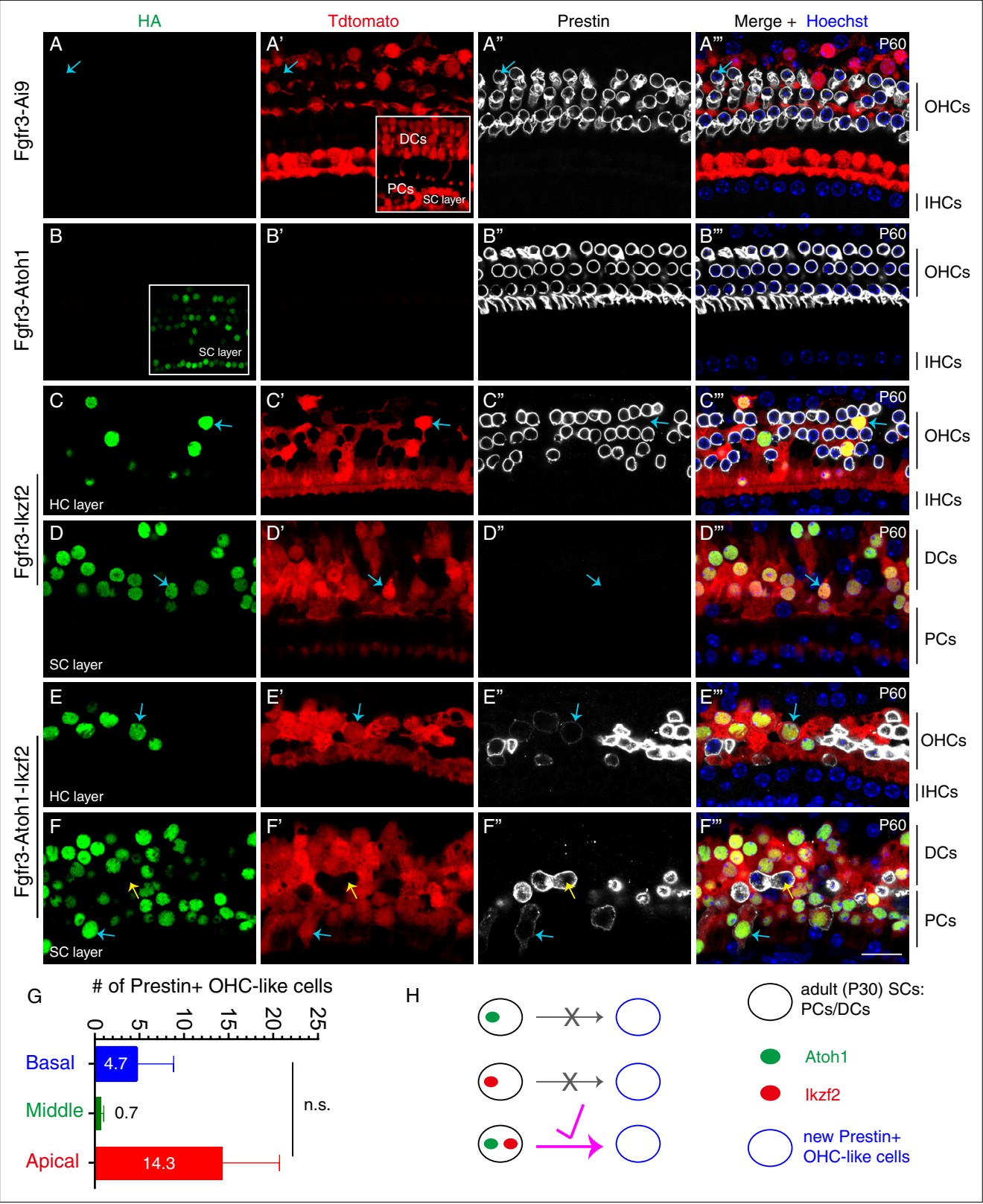

**Figure 2.** Joint expression of Atoh1 and Ikzf2 in adult SCs (PCs and DCs) induced transition to an OHC-like cell state at low frequency. Triple labeling against HA, Tdtomato, and Prestin in four different mouse genetic models that were administered tamoxifen at P30 and P31 and sacrificed at P60. Confocal images scanned at HC or SC layers. (**A–A'''**) In the control Fgfr3-iCreER+; Ai9/+ (Fgfr3-Ai9) mice, Tdtomato labelled SCs (primarily PCs and DCs) and no Prestin expression was observed in them. Inset in (**A'**): scan at SC layer. Arrows in (**A–A'''**): Prestin+/Tdtomato- OHC. (**B–B'''**) In mice

*Figure 2 continued on next page*

*Figure 2 continued*

expressing Atoh1 in adult SCs, Fgfr3-iCreER+; CAG-LSL-*Atoh1*+ (Fgfr3-Atoh1), no Tdtomato signal was detected and no HA+/Prestin+ cells were observed. Inset in (**B**): scan at SC layer. (**C–D'''**) In mice expressing Ikzf2 in adult SCs, Fgfr3-iCreER+; *Rosa26*^CAG-LSL-Ikzf2/+^ (Fgfr3-Ikzf2), HA and Tdtomato were co-expressed in both the HC (**C–C'''**) and SC (**D–D'''**) layer. Arrows in both layers: two cells that were HA+/Tdtomato + but did not express Prestin. Alignment of endogenous Prestin+ OHCs was abnormal and some of them were lost. (**E–F'''**) In mice, Fgfr3-iCreER+; CAG-LSL-*Atoh1*+; *Rosa26*^CAG-LSL-Ikzf2/+^ mice (Fgfr3-Atoh1-Ikzf2), some adult SCs were HA+/Tdtomato+/Prestin+, taking on an OHC-like phenotype. Arrows in (**E–E'''**) indicated an OHC-like cell originating from an adult DC; in contrast, the blue arrows in (**F–F'''**) indicated another OHC-like cell derived from adult PCs. Yellow arrows in (**F–F'''**): Prestin+/Tdtomato- endogenous OHC appearing in the SC layer. Prestin expression in the new OHC-like cells was noted to be lower than that in wild-type endogenous OHCs. (**G**) Quantification of the number of OHC-like cells throughout the cochlear turns in the Fgfr3-Atoh1-Ikzf2 mice. Data was presented as mean ± SEM (n = 3); no statistical difference was detected between regions (Student's t test). OHC-like cells were reliably observed in all animals, but cell numbers were low and showed large variations. (**H**) Summary of reprogramming outcomes in the three mice models. OHC-like cells were present only when both Atoh1 and Ikzf2 were concurrently reactivated in adult cochlear SCs. Scale bar: 20 µm.

The online version of this article includes the following figure supplement(s) for figure 2:

**Source data 1.** It contained the numerical values of the data plotted in the graph of *Figure 2G*.

albeit at low efficiency, adult cochlear SCs into Prestin+ OHC -like cells, whilst independently, neither gene was able to trigger this conversion (*Figure 2H*). In an attempt to boost the reprogramming efficiency, we next sought to test whether pre-damaging OHCs might act as a permissive and promoting factor to SCs conversion into Prestin+ OHC -like cells.

## Generation of *Slc26a5*^DTR/+^ mouse line to study effects of OHC damage

To non-invasively damage adult OHCs in vivo, we resorted to making use of the diphtheria toxin (DT)/ DT receptor (DTR) system, which has already been used to successfully lesion HCs in the inner ear (*Cox et al., 2014*; *Golub et al., 2012*; *Tong et al., 2015*). We generated a new knock-in mouse model, *Slc26a5*-P2A-DTR/+ (*Slc26a5*^DTR/+^ for short), in which the P2A-DTR fragment was inserted immediately before the stop codon (TAA) of *Slc26a5* (enconding the OHC marker, Prestin) (*Figure 3—figure supplement 1A–C*). As a result DTR expression was entirely controlled by the endogenous *Slc26a5* promoter and/or enhancers and thus restricted to being expressed in OHCs, whilst still leaving OHC Prestin expression itself in principle unaffected. Southern blotting results confirmed the absence of random insertion of donor DNA in the host genome *Figure 3—figure supplement 1D,E*, and tail-DNA PCR allowed us to readily distinguish between wild-type and post-gene-targeting *Slc26a5* alleles (*Figure 3—figure supplement 1F*).

In the absence of DT treatment, co-staining for the OHC-marker, Prestin and IHC-marker, vGlut3 revealed that OHCs were normal in P42 *Slc26a5*^DTR/+^ mice (n = 3) (*Figure 3A and A'*). In contrast, after a single injection of DT (20 ng/g, body weight) at P36, severe OHC loss was observed in *Slc26a5*^DTR/+^ mice (n = 3) at P42, with only a few OHCs detectable throughout the cochlear duct (arrowhead in *Figure 3B'*). Debris from dying OHCs were frequently observed at P42 (arrows in *Figure 3B'*) but had disappeared by P60 (see below for details; see *Figure 4*). Conversely, IHCs appeared normal at P42 under DT treatment, which confirmed that DTR was indeed specifically expressed in OHCs only (*Figure 3B and B'*). Furthermore, the results of auditory brainstem response (ABR) measurement demonstrated that auditory thresholds at distinct frequencies in *Slc26a5*^DTR/+^ mice (n = 3) treated with DT were significantly higher than those in untreated control *Slc26a5*^DTR/+^ mice (n = 3) (*Figure 3C*). In logical accord with this, the ratio of OHC numbers to IHC number was significantly reduced in *Slc26a5*^DTR/+^ mice treated with DT (red in *Figure 3D*) being at ~0.19, compared to the ratio of ~3.23 in control population (blue in *Figure 3D*). Together, these results showed that DT treatment caused, within 6 days, marked hearing impairment due to selective OHC loss, and validated the use of *Slc26a5*^DTR/+^ as a powerful mouse model to study the selective effects wild-type endogenous OHC loss. We also note that, thanks to the exclusive expression of Prestin in cochlear OHCs, *Slc26a5*^DTR/+^ mice displayed no detectable abnormalities, apart from the desired deafness following DT treatment.

## Endogenous OHC damage considerably enhances capacity of Ikzf2 and Atoh1 to convert adult cochlear SCs into Prestin+ OHC-like cells

In non-mammalian vertebrates, HC loss triggers the regeneration of HCs through a cell-fate change in SCs (*Janesick and Heller, 2019*; *Stone and Cotanche, 2007*; *Warchol and Corwin, 1996*). This prompted us to test whether damaging endogenous wild-type adult OHCs would engender a

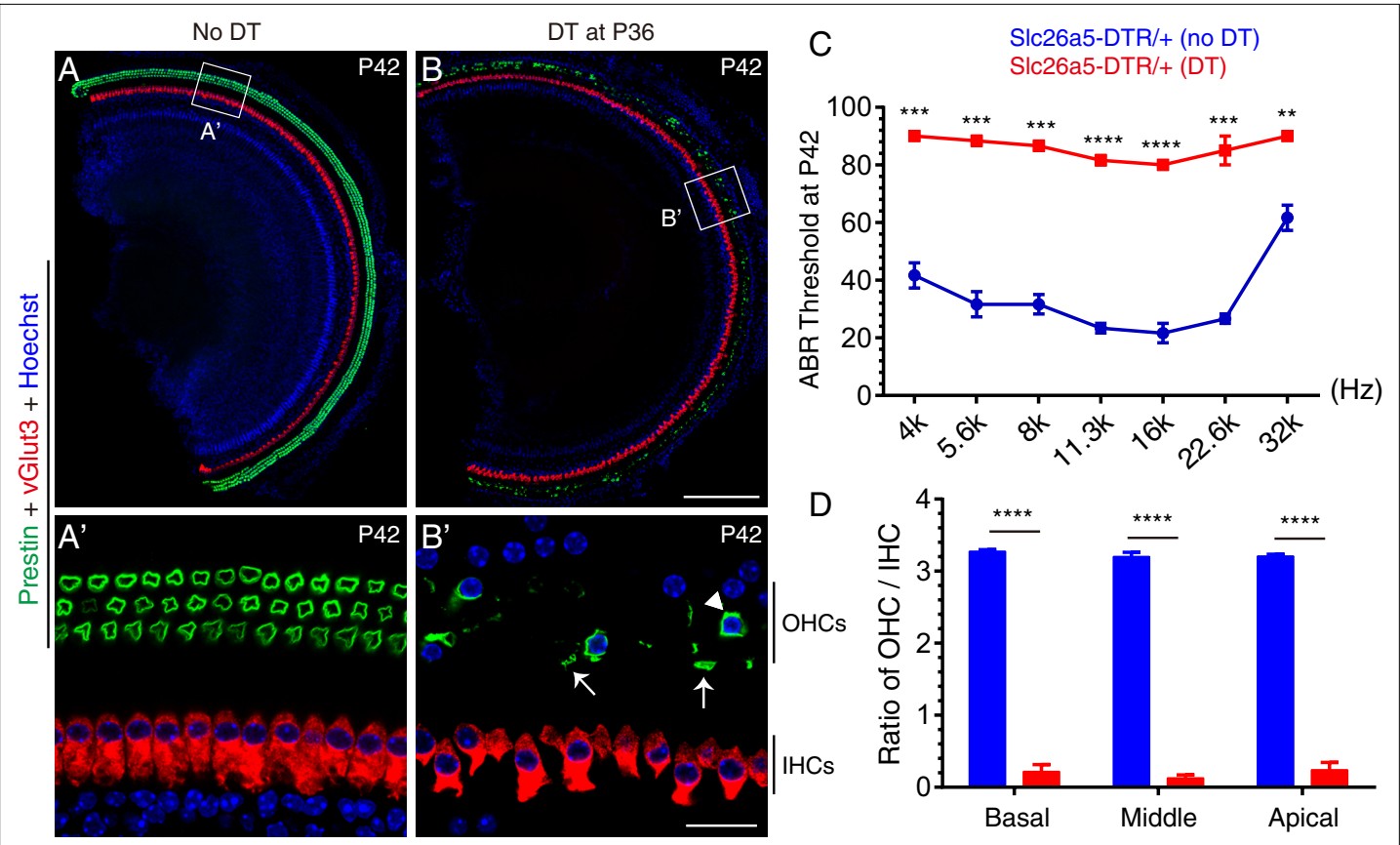

**Figure 3.** Targeted lesioning of adult endogenous OHCs via diphtheria toxin treatment. (**A–B'**) *Slc26a5*^DTR/+ mice were treated without (**A–A'**, control) or with (**B-B'**, experimental) diphtheria toxin (DT) at P36 and analyzed at P42. Samples were co-stained against Prestin and vGlut3. (**A'**) and (**B'**): magnified images of boxed region in (**A**) and (**B**). DT treatment led to rapid OHC death within 6 days; only a few OHCs remained (arrowhead in B'); debris of dying OHCs were frequently detected (arrows in B'). Much of the green signal in (**B**) came from the dying OHC debris. (**C**) Auditory brainstem response (ABR) threshold measurement of *Slc26a5*^DTR/+ mice treated with (red line) or without (blue line) DT. After DT treatment ABR thresholds were significantly higher throughout the frequency ranges. (**D**) Ratios of OHCs to IHCs within a same scanning area in control mice (blue) and experimental mice (red); relative OHC numbers were significantly reduced in the experimental mice. Data was presented as mean ± SEM (n = 3). Student's t-test were run for statistical testing. **p<0.01, ***p<0.001, ****P<0.0001. Scale bars: 200 μm (**B**), 20 μm (**B'**).

The online version of this article includes the following figure supplement(s) for figure 3:

**Source data 1.** It contained the numerical values of the ABR thresholds presented in *Figure 3C*.

**Source data 2.** It contained the numerical value of the ratio of OHCs to IHCs described in *Figure 3D*.

**Figure supplement 1.** Generation of the *Slc26a5*^DTR/+ knockin mouse model.

more favorable environment for SC transdifferentiation, thereby potentially increasing the conversion efficiency of adult cochlear SCs overexpressing Ikzf2 and Atoh1 into OHC-like cells. For this, we compared four genetic mouse strains: (1) *Slc26a5*^DTR/+; (2) Fgfr3-iCreER+; CAG-LSL-*Atoh1*+; *Slc26a5*^DTR/+ (Fgfr3-Atoh1-DTR); (3) Fgfr3-iCreER+; *Rosa26*^CAG-LSL-Ikzf2/+; *Slc26a5*^DTR/+ (Fgfr3-Ikzf2-DTR); and (4) Fgfr3-iCreER+; CAG-LSL-*Atoh1*+; *Rosa26*^CAG-LSL-Ikzf2/+; *Slc26a5*^DTR/+ (Fgfr3-Atoh1-Ikzf2-DTR). All mice were first injected tamoxifen at P30 and P31, to trigger expression of Atoh1 or Ikzf2 or both or non. Six days later, at P36, DT was administered to trigger OHC damage (*Figure 4A,B*). By injecting Tamoxifen, prior to DT, we could selectively and permanently label adult SCs (mainly PCs and DCs) with HA and Tdtomato before any OHC damage. The reverse order was not used so as to avoid the possibility of OHC damage leading to changes in the Cre-expression pattern in Fgfr3-iCreER+ mice.

Akin to before, P60 *Slc26a5*^DTR/+ mice treated with DT had few visible OHCs (arrows in *Figure 4D*) as compared to the untreated, same aged, *Slc26a5*^DTR/+ control mice (*Figure 4C*). However, unlike the findings at P42, at P60 little OHC debris was observed. Furthermore, HA+/Prestin+ cells were identified in neither Fgfr3-Atoh1-DTR mice (*Figure 4E*) nor Fgfr3-Ikzf2-DTR mice (*Figure 4F*) at P60,

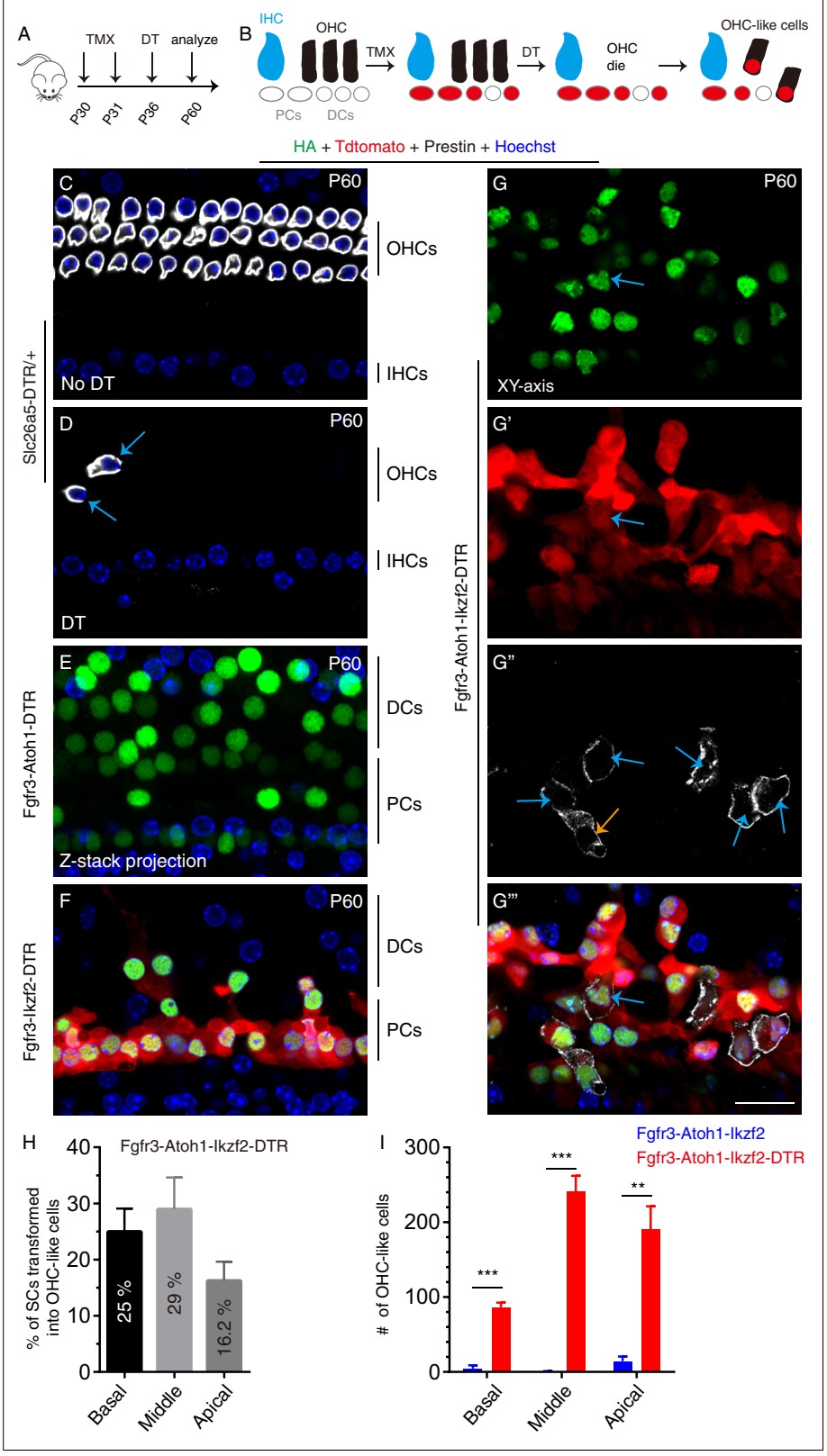

**Figure 4.** Lesion to endogenous OHCs enhances reprogramming efficiency of adult SCs by Atoh1 and Ikzf2. (**A**) Three distinct mice models were subject to tamoxifen (TMX) treatment at P30 and P31, followed by DT treatment at P36 and analysis at P60. (**B**) Illustration of key cellular events occurring during the experiment: induction of Atoh1 and Ikzf2 in adult PCs and DCs, which were permanently labeled with Tdtomato for the subsequent fate-mapping

*Figure 4 continued on next page*

*Figure 4 continued*

analysis. (**C–G'''**) Triple labeling against HA, Tdtomato, and Prestin in four models: (1) *Slc26a5^{DTR/+}* (**C, D**), (2) Fgfr3-iCreER+; CAG-LSL-*Atoh1*+; *Slc26a5^{DTR/+}* (Fgfr3-Atoh1-DTR; **E**), (3) Fgfr3-iCreER+; *Rosa26^{CAG-LSL-Ikzf2/+}*; *Slc26a5^{DTR/+}* (Fgfr3-Ikzf2-DTR; **F**), and (4) Fgfr3-iCreER+; CAG-LSL-*Atoh1*+; *Rosa26^{CAG-LSL-Ikzf2/+}*; *Slc26a5^{DTR/+}* (Fgfr3-Atoh1-Ikzf2-DTR; **G–G'''**). Relative to *Slc26a5^{DTR/+}* mice not treated with DT (**C**), DT-treated *Slc26a5^{DTR/+}* mice harbored fewer normal Prestin+ OHCs at P60 (arrows in **D**). Debris of dying OHCs had disappeared. No OHC-like cells were observed in the first three models, but Tdtomato+/HA+/Prestin+ OHC -like cells (orange and blue arrows in **G''**) were present in the Fgfr3-Atoh1-Ikzf2-DTR model (**G-G'''**). (**H**) Percentages of SCs that were transformed into OHC-like cells at different cochlear turns in Fgfr3-Atoh1-Ikzf2-DTR mice. (**I**) Comparison of OHC-like cell numbers between Fgfr3-Atoh1-Ikzf2-DTR and Fgfr3-Atoh1-Ikzf2 models (without damaging adult wild-type OHCs). Fgfr3-Atoh1-Ikzf2-DTR mice harbored considerably more OHC-like cells than Fgfr3-Atoh1-Ikzf2 mice. Data was presented as mean ± SEM (n = 3). Student's t test was used for statistical analysis. **p<0.01, ***p<0.001. Scale bar: 20 μm.

The online version of this article includes the following figure supplement(s) for figure 4:

**Source data 1.** It contained the numerical values of the data plotted in the graph of *Figure 4H*.

**Source data 2.** It contained the numerical values of the data plotted in *Figure 4I*.

**Figure supplement 1.** Visualization of OHC-like cells in orthogonal projections.

**Figure supplement 2.** Nascent HCs emerged at P42 and OHC-like cells at P46.

**Figure supplement 2—source data 1.** It contained the number of new HCs at P42 and P46 presented in the graph of *Figure 4—figure supplement 2D*.

**Figure supplement 2—source data 2.** It contained the numbers of new HCs at P46 presented in the graph of *Figure 4—figure supplement 2E*.

suggesting that Atoh1 or Ikzf2 alone was insufficient to convert mature SCs into OHC-like cells after endogenous OHC damage.

In contrast, in P60 Fgfr3-Atoh1-Ikzf2-DTR mice, Tdtomato+/HA+/Prestin+ OHC -like cells were frequently observed. Confocal scanning of the entire cochlear duct (n = 3) revealed the presence of 359.3 ± 46.2, 878.0 ± 118.7, and 1195 ± 81.6 Tdtomato+/HA+ cells of which 86.3 ± 6.3, 241.0 ± 21.1, and 190.3 ± 31.1 cells were new Prestin+ OHC -like cells (arrows in *Figure 4G–G'''*, *Figure 4—figure supplement 1A–B'''* and *Video 1*) in the basal, middle, and apical turns, respectively. Or in other words, 25.0% ± 4.1%, 29.0% ± 5.7%, and 16.2% ± 3.4% of adult cochlear tdTomato+ SCs trans-differentiated into OHC-like cells in the basal, middle, and apical turns, respectively (*Figure 4H*). This rate was significantly greater (*Figure 4I*) than that obtained with un-lesioned P60 Fgfr3-Atoh1-Ikzf2 mice (*Figure 2E–G*). Expression levels of Prestin were also higher in mice with pre-lesioned OHCs (*Figure 4G–G'''*), but still remained considerably lower than those in wild-type endogenous OHCs (*Figure 4C* vs *Figure 4G''*). In addition, ~ 82.6 % of OHC-like cells were located in the top HC layer (blue arrows in *Figure 4G''*, *Figure 4—figure supplement 1A–A'''*), whereas the rest remained in the bottom SC layer (orange arrows in *Figure 4G''*, *Figure 4—figure supplement 1B–B'''*). These results demonstrated that damaging endogenous OHCs markedly enhanced the reprogramming ability of Ikzf2 and Atoh1 to convert adult cochlear SCs into OHC-like cells.

## The conversion of SCs to OHC-like cells involves an intermediate general HC state

We next determined the time it took for adult cochlear SCs to form nascent HCs and OHC-like cells (*Figure 4—figure supplement 2A–C'''*). Nascent HCs were defined as Tdtomato+/Myo6+/Prestin− (arrowheads in *Figure 4—figure supplement 2B–B'''*). OHC-like cells were defined as

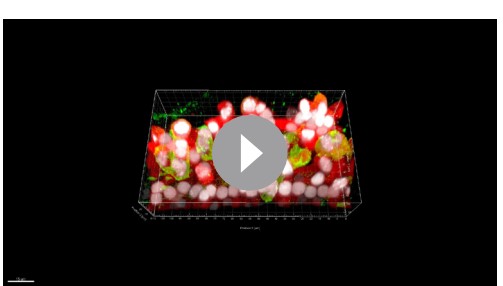

**Video 1.** The confocal data presented in Figure 4G–G''' and Figure 4—figure supplement 1A–A''' was converted to Imaris format and loaded in the Imaris software to more readily visualize OHC-like cells using orthogonal projections of the data. Red, green, and white colors represented Tdtomato signal and Prestin and HA labelling respectively. OHC-like cells were defined as triple-positive cells.

https://elifesciences.org/articles/66547/figures#video1

Tdtomato+/Myo6+/Prestin+. P42 was the earliest age at which the nascent HCs were detected; Myo6 expression was weak at this stage. Scanning the entire cochlear duct at P42 revealed the presence of only 29.0 ± 13.5, 66.0 ± 35.23, and 52.7 ± 25.4 nascent HCs throughout the basal, middle, and apical turns (n = 3), respectively (black in *Figure 4—figure supplement 2D*). No Tdtomato+/Myo6+/ Prestin+ OHC- like cells were detected at P42.

Four days later, at P46 (n = 4), there were 56.5 ± 28.8, 118.0 ± 61.2, and 111.0 ± 57.3 new HCs that were Tdtomato+/Myo6+ (gray in *Figure 4—figure supplement 2D,F*), and, of these, 27.8 ± 15.0, 43.5 ± 25.3, and 36.8 ± 21.0 were Tdtomato+/Myo6+/Prestin+ (classified as OHC-like cells; green in *Figure 4—figure supplement 2E*); thus, OHC-like cells accounted for 49.2 % (27.8/56.5), 36.9 % (43.5/118), and 33.2 % (36.8/111) of total new HCs in basal, middle, and apical turns, respectively. As mentioned, the remaining Tdtomato+/Myo6+/Prestin− cells were defined as nascent HCs, and the Tdtomato+/Myo6-/Prestin− cells were defined as 'SCs failing to become HCs'. Notably, we sorted new HCs into nascent HCs and OHC-like cells based solely on absence or presence of Prestin. Among the four mice analyzed, two mice harbored substantially fewer Tdtomato+/Myo6+ cells than the others, which caused the large variations in numbers; nevertheless, the overall trend was that the larger the number of Tdtomato+/Myo6+ cells, the larger the number of Tdtomato+/Myo6+/Prestin+ OHC -like cells. Importantly, no Tdtomato+/Prestin+ cells lacking Myo6 expression were detected. Together, these findings suggested that the generation of new OHC-like cells generally involved an initial cell-fate transition from SCs (PCs and DCs) into nascent HCs, which occurred by P42 (12 days

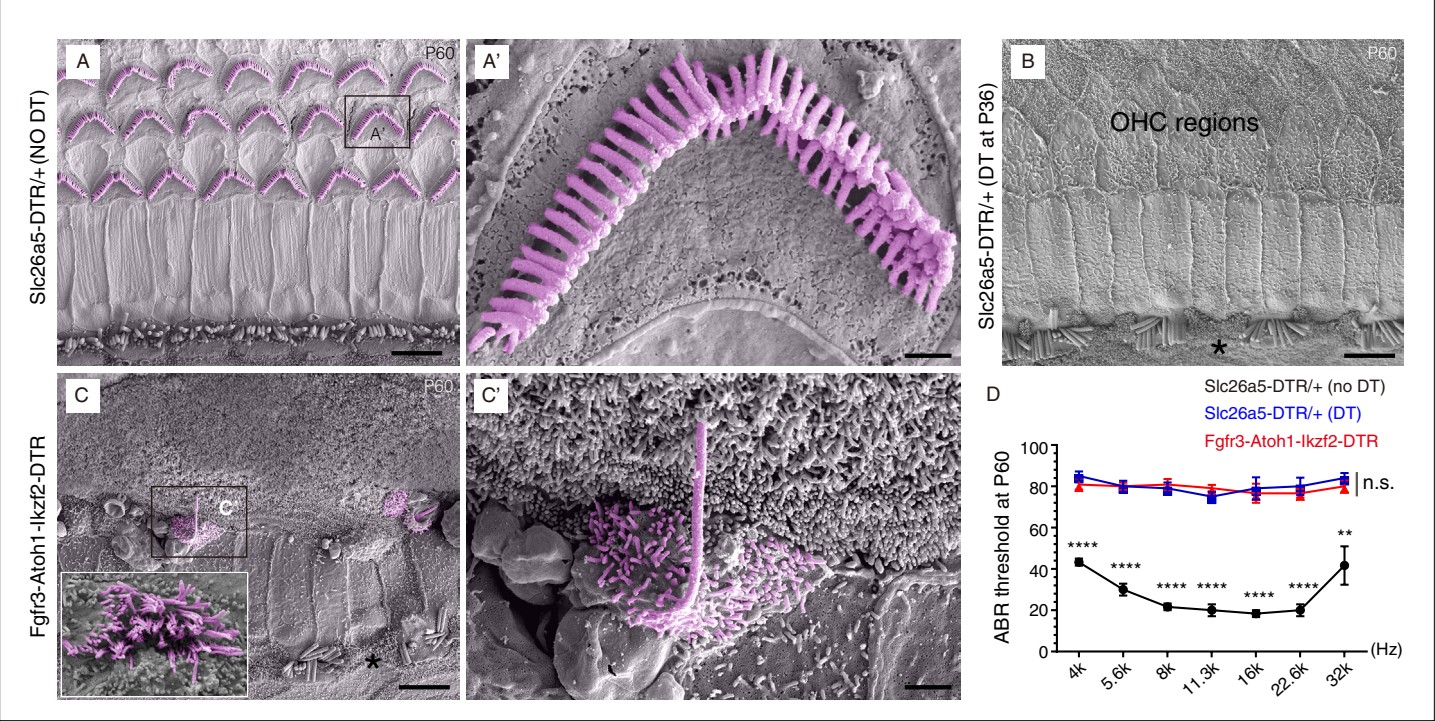

**Figure 5.** Hair bundles were present in OHC-like cells. Scanning electron microscopy (SEM) analysis of samples from three different mouse models at P60. (**A–A'**) OHCs harbored V- or W-shaped hair bundles in S*lc26a5*[DTR/+] mice not treated with DT. (**A'**): high-magnification view of boxed region in (**A**). (**B**) Majority of OHCs had disappeared by P60 in *Slc26a5*[DTR/+] mice upon DT treatment at P36. Black asterisk: one IHC that was absent. (**C**) Immature hair bundles were frequently detected in the Fgfr3-iCreER+; CAG-LSL-*Atoh1*+; Rosa26[CAG-LSL-Ikzf2/+]; *Slc26a5*[DTR/+] (Fgfr3-Atoh1-Ikzf2-DTR) model, but not in (**A**) and (**B**). These hair bundles were thought to originate from OHC-like cells. (**C'**): high-magnification view of boxed region in (**C**). Inset in C: instance of occasional hair bundles. Black asterisk: one IHC that was missing. (**D**) ABR measurements of the three mice models. Relative to the untreated *Slc26a5*[DTR/+] control mice (black line, n = 3), the ABR thresholds of *Slc26a5*[DTR/+] treated with DT (blue line, n = 5) and in Fgfr3-Atoh1-Ikzf2-DTR mice (red line, n = 6) were significantly higher. The blue and red lines showed no statistically significant difference at any frequency (n.s.). Data was presented as mean ± SEM. Student's t-test was used for statistical testing. **p<0.01, ****p<0.0001. Scale bars: 5 μm (**A–C**), 1 μm (**C'**), and 500 nm (**A'**).

The online version of this article includes the following figure supplement(s) for figure 5:

**Source data 1.** It contained the numerical values of the ABR thresholds described in *Figure 5D*.

after Atoh1 and Ikzf2 gene induction, and 6 days after DT treatment), and that this was followed by a second transition from nascent HCs into OHC-like cells.

## New OHC-like cells are able to produce hair bundles

Scanning electron microscopy (SEM) was used to ascertain the presence and morphology of hair bundles (a cluster of stereocilia) of the Prestin+ new OHC-like cells at P60 (*Figure 5*). These characteristic staircase-shaped hair bundles are where mechanoelectrical transduction (MET) channels critical for hearing reside (*Corey and Holt, 2016*; *Douguet and Honore, 2019*; *Wu and Muller, 2016*). In non-DT-treated control *Slc26a5^DTR/+* mice, the stereotyped V- or W-shaped hair bundles were present in OHCs (*Figure 5A,A′*), whereas very few hair bundles remained in *Slc26a5^DTR/+* mice treated with DT (*Figure 5B*). This was consistent with the results from the immunostaining assays (*Figure 4D*).

Intriguingly, in Fgfr3-Atoh1-Ikzf2-DTR P60 mice, we frequently detected stereocilia with a single long bundle, lacking the staircase shape (*Figure 5C–C′*). Stereocilia with several long bundles were seldom observed (inset in *Figure 5C*). These rare hair bundle likely belonged to OHC-like cells (or nascent HCs), as such hair bundles were never observed in *Slc26a5^DTR/+* mice, irrespective of DT treatment (*Figure 5A,B*). Auditory brain response (ABR) measurement showed that the thresholds at distinct frequencies were markedly higher in Fgfr3-Atoh1-Ikzf2-DTR mice (n = 6, red line in *Figure 5D*) compared to non-DT-treated *Slc26a5^DTR/+* mice (n = 3, black line in *Figure 5D*). Regretfully, no hearing improvement (lowering of threshold) was apparent between Fgfr3-Atoh1-Ikzf2-DTR compared to DT-treated *Slc26a5^DTR/+* mice (n = 5, blue line in *Figure 5D*). Notably, we observed IHC loss at P60 (asterisks in *Figure 5B,C*). This could be a secondary effect of OHC damage, since no IHC death was observed at P42 (*Figure 3*). Collectively, these results showed that the new OHC-like cells still differed from adult endogenous OHCs in terms of hair-bundle abundance and morphology as well as Prestin expression levels, and were not yet functional. The degree to which OHC-like cells resembled adult endogenous OHCs was investigated next through single-cell RNA-Seq analyses.

## Single-cell RNA-Seq reveals genes selectively enriched in adult wild-type OHCs and SCs

To perform single-cell RNA-Seq on adult OHCs and SCs, we manually picked 17 wild-type Tdtomato+ OHCs from *Slc26a5^CreER/+*; Ai9/+ mice at P30 and 16 wild-type Tdtomato+ SCs (primarily PCs and DCs) from Fgfr3-iCreER+; Ai9/+ mice at P60 (*Figure 6A*). *Slc26a5^CreER/+* is a knock-in mouse strain with OHC-specific Cre activity (*Fang et al., 2012*). All cells were identified based on their endogenous Tdtomato fluorescence and were picked and washed thrice under a fluorescence microscope before final placement in PCR tubes. RNA-Seq libraries were then prepared using the Smart-Seq approach (*Figure 6A*). Manual picking combined with Smart-Seq has been successfully used in previous gene-profiling studies on adult cochlear HCs and SGNs (*Li et al., 2020b*; *Liu et al., 2014b*; *Shrestha et al., 2018*).

We first compared gene profiles between adult endogenous OHCs and SCs, and identified 1051 and 1982 genes enriched respectively (p<0.001) in adult OHCs and SCs (*Figure 6—figure supplement 1A*); *Supplementary file 1* contained the full list of enriched genes. The list of these adult OHC-enriched genes (OHC genes for short) included previously known highly expressed adult OHC genes, such as *Myo7a, Ocm, Slc26a5* (Prestin), *Ikzf2, Espn, Tmc1, Cib2, Lhfpl5, Lmo7, Lbh,* and *Sri* (*Chessum et al., 2018*; *Du et al., 2019*; *Giese et al., 2017*; *Liu et al., 2014b*; *Ranum et al., 2019*; *Xiong et al., 2012*; *Zheng et al., 2000*). Similarly, previously known pan-SC markers, such as *Sox2* and *Sox10*, and two recently identified DC-specific genes, *Bace2* and *Ceacam16* (*Li et al., 2018b*; *Ranum et al., 2019*), were found amongst the adult SC-enriched genes (SC genes for short). Gene Ontology (GO) analysis showed that OHC genes were involved in sensory perception of sound, inner ear morphogenesis and stereocilium organization (*Figure 6—figure supplement 1B*). OHC genes associated with each GO category were summarized in *Supplementary file 2*. These results showed that the picked adult endogenous OHCs and SCs were pure and that our single cell RNA-Seq data was of high quality. These differentially enriched genes identified in OHCs and SCs were subsequently used as reference to assess the degree of cell fate conversion of SC-derived nascent HCs and OHC-like cells.

Some unexpected findings from the RNA-Seq data included *Myo6* expression levels: although Myo6 protein is known to be enriched in OHCs, *Myo6* mRNA was not significantly enriched in adult OHCs, likely because *Myo6* mRNA was also detected (albeit at a lower level) in adult SCs. *Myo6*

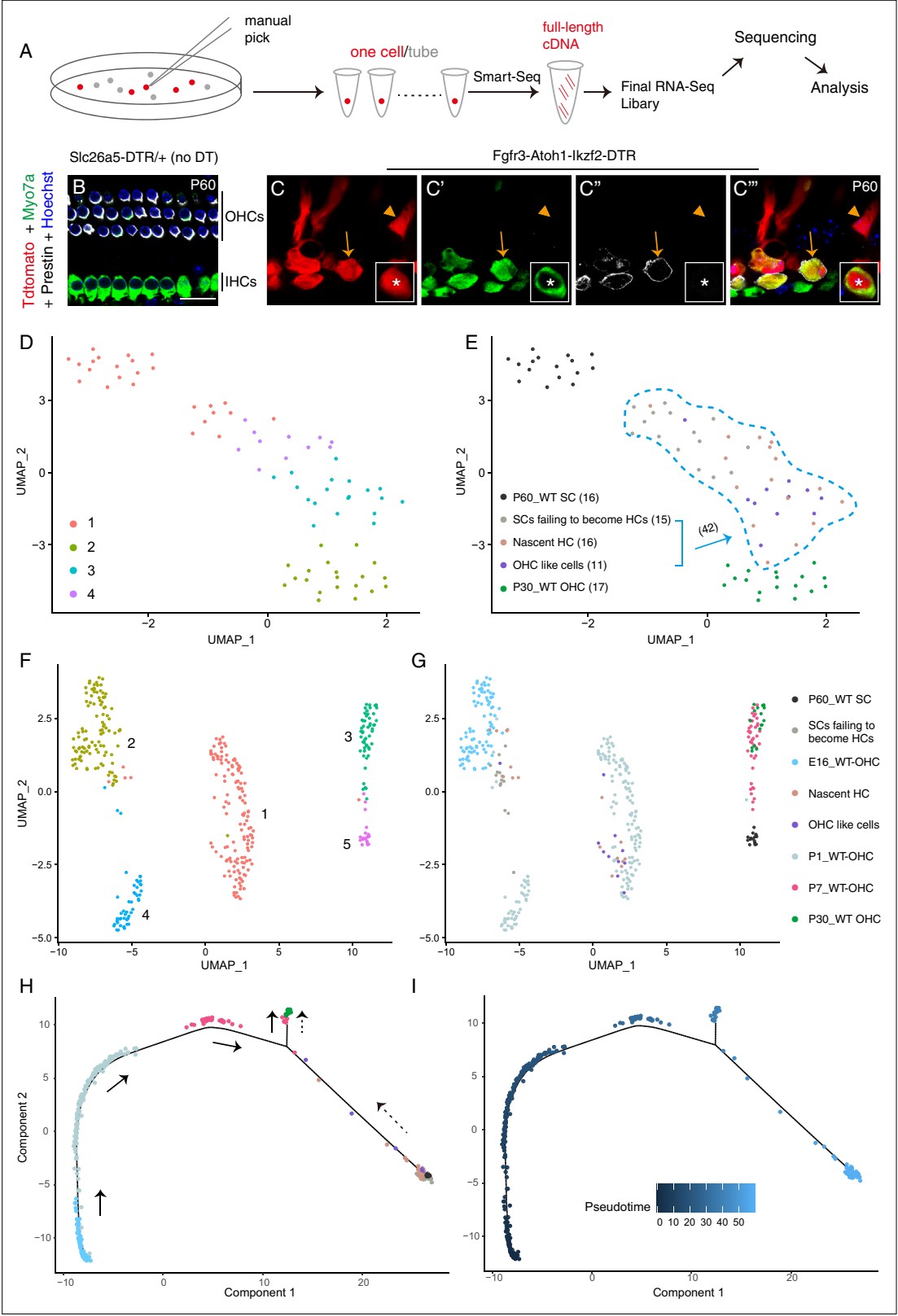

**Figure 6.** OHC-like cells were most akin to ~ P1 wild-type OHCs. (**A**) Illustration of the single cell RNA-Seq experimental pipeline. (**B–C′′′**) Triple labeling against Myo7a (maker for HCs), Prestin, and Tdtomato in cochlear samples from control (**B**) and Fgfr3-Atoh1-Ikzf2-DTR (**C**) mice at P60. Arrows: Tdtomato+/Myo7a+/Prestin+ OHC -like cells; arrowheads: Tdtomato+/Myo7a-/Prestin− cell, defined as SCs failing to become HCs; asterisks: nascent HC that was Tdtomato+/Myo7a+/Prestin− (inset in **C–C′′′**). (**D–E**) UMAP embedding of RNA-seq data revealed four main clusters (**D**). The same plot

*Figure 6 continued on next page*

*Figure 6 continued*

was presented with the five different cell types included in the data labelled separately (**E**). The 42 Tdtomato+ cells within the light-blue dotted lines (**E**) originated from the P60 Fgfr3-Atoh1-Ikzf2-DTR mice. When unsupervised clustering was performed over all 42 Tdtomato+ cells, three clusters were identified. (**F–G**) UMAP analysis of cells in (**D–E**) as well as wild-type OHCs at E16, P1, and P7 (E16_WT OHCs, P1_WT OHCs, and P7_WT OHCs; further information was presented in *Figure 6—figure supplement 5*). OHC-like cells were overlapped with P1_WT OHCs. (**H–I**) Trajectory analysis of all cells in (**F–G**). Black arrows in (**H**) represented the developmental trajectory of endogenous OHCs between E16 and P30, whilst the black dotted arrows marked the reprogramming trajectory from endogenous adult SCs to endogenous adult OHCs. Scale bar: 20 µm.

The online version of this article includes the following figure supplement(s) for figure 6:

**Figure supplement 1.** Transcriptomic comparison between adult wild-type OHCs and SCs.

**Figure supplement 2.** OHC-like cells upregulated 729 OHC genes and downregulated 331 SC genes.

**Figure supplement 3.** Two trajectory analyses by different cell pooling.

**Figure supplement 4.** Heterogeneous expression of OHC genes across OHC-like cells.

**Figure supplement 5.** Trajectory analysis of wild type differentiating OHCs.

**Figure supplement 6.** OHC-like cells primarily expressed neonatal OHC, but minimally expressed neonatal IHC or utricle HC genes.

**Figure supplement 7.** Transcriptomic differences between OHC-like cells and mature OHCs.

mRNA is also known to be present in other non-HC populations (*Kolla et al., 2020*; *Scheffer et al., 2015*). Also noteworthy, *Myo7a* was significantly enriched in adult OHCs (red arrow in *Figure 6—figure supplement 1A*) and also highly expressed in nascent HCs and OHC-like cells; however, this was not the case in SCs failing to become HCs (*Figure 6B–C‴*). This led us to use *Myo7a* as an early HC marker and as a means of defining general HC fate in the RNA-Seq analysis described below.

## OHC-like cells upregulate OHC genes and downregulate SC genes

We next focused on comparing global gene profiles between OHC-like cells, adult endogenous OHCs, and SCs. We manually picked 42 Tdtomato+ cells from Fgfr3-Atoh1-Ikzf2-DTR mice at P60 (*Figure 6A*) and sorted them into three types: (1) Tdtomato+/Myo7a+/Prestin+ cells (defined as OHC-like cells, arrows in *Figure 6C–C‴*; n = 11 cells); (2) Tdtomato+/Myo7a+/Prestin– cells (defined as nascent HCs, asterisks in inset of *Figure 6C–C‴*; n = 16); and (3) Tdtomato+/Myo7a-/Prestin– cells (defined as SCs failing to become HCs, arrowheads in *Figure 6C–C‴*; n = 15). The proportion of OHC-like cells was 26.2 % (11/42), which agreed with the immunostaining results (16.2%–29.0%) (*Figure 4H*), giving us confidence in the suitability of this criterion. Moreover, expression of *Atoh1* and *Ikzf2* (arrows in *Figure 6—figure supplement 2A*) was enriched in all 42 cells, but not in adult endogenous SCs. This also confirmed that Atoh1 and Ikzf2 were permanently overexpressed in the 42 Tdtomato+ cells, regardless of cell fate.

UMAP (uniform manifold approximation and projection) analysis demonstrated that OHC-like cells differed more greatly from adult endogenous P60 SCs (hereafter named as WT_P60 SCs) and were more similar to P30 adult endogenous OHCs (hereafter named as WT_P30 OHCs) than SCs failing to become HCs (*Figure 6D,E*). Besides *Slc26a5* (Prestin) and *Myo7a* (purple arrows in *Figure 6—figure supplement 2A*), 1313 genes were expressed at a significantly higher level in OHC-like cells relative to WT_P60 SCs. GO analysis of these 1313 genes revealed that they were associated with functions involving sensory perception of sound (*Figure 6—figure supplement 2B*), which further supported the notion that OHC-like cells globally behave like HCs. *Supplementary file 3* included all genes associated with the GO analysis. Notably, 55.5 % (729/1313) of the genes expressed more in OHC-like cells than WT_P60 SCs overlapped with OHC genes, such as *Myo7a*, *Pvalb*, *Calb1*, *Cib2*, and *Lhfpl5* (purple arrows in *Figure 6—figure supplement 2A*). Conversely, 527 genes were expressed at significantly lower levels in OHC-like cells as compared to WT_P60 SCs, of which 62.8 % (331/527) overlapped with SC genes, such as *Tuba1b*, *Gjb2*, and *Rorb* (black arrows in *Figure 6—figure supplement 2A*). *Supplementary file 4* contained the entire list of genes that were differently expressed between OHC-like cells and WT_P60 SCs, as well as the 729 OHC genes that were upregulated, and 331 SC genes that were down-regulated in OHC-like cells. Furthermore, we also included data from nascent HCs and SCs failing to become HCs for completeness and as potential intermediate cell types. In general, the identified genes above were either not or only slightly upregulated or downregulated in SCs failing to become HCs (*Figure 6—figure supplement 2A*). Together, this suggested that OHC-like cells were on an OHC-lineage differentiation track.

To our surprise, trajectory analysis including the 42 Tdtomato+ cells suggested that OHC-like cells and nascent HCs were closer to each other, relative to SCs failing to become HCs (*Figure 6— figure supplement 3A,B*). Furthermore, another trajectory analysis confirmed that OHC-like cells and nascent HCs were closer to WT_P30 OHCs, and conversely SCs failing to become HCs were closer to WT_P60 SCs (*Figure 6—figure supplement 3C,D*). The transcriptomic similarity between OHC-like cells and nascent HCs supported that, despite Prestin expression only being present in OHC-like cells, the general expression patterns of other HC genes might be quite similar. This possibility was partially supported by the finding that not all Prestin+ OHC -like cells expressed the early pan-HC markers, Rbm24, Pvalb, and Calb1, according to both RNA-Seq and immunostaining results (*Figure 6—figure supplements 2 and 4*). These pan-HC markers are typically expressed earlier than Prestin in endogenous OHCs (*Grifone et al., 2018*; *Li et al., 2018b*; *Liu et al., 2012a*; *Wang et al., 2021*). Together, our results showed that OHC-like cells/nascent HCs resembled WT_P30 OHCs considerably more than SCs failing to become HCs. Next, we evaluated the differentiation status of the OHC-like cells.

## OHC-like cells resemble wild-type neonatal differentiating OHCs

By reanalyzing previously published data from a recent single-cell RNA-Seq studies covering wild-type cochlear OHCs, IHCs, SCs, GER (greater epithelial ridge) cells, and LER (lesser epithelial ridge) cells (*Kolla et al., 2020*), we could compare OHC-like cell differentiation status at finer granularity. More specifically, we extracted from these data OHCs at three different ages: E16 (n = 87), P1 (n = 170), and P7 (n = 39) (*Figure 6—figure supplement 5A*). These pooled OHCs were highly heterogeneous, especially at P1, due in part to the basal-to-apical and medial-to-lateral developmental gradient in the cochlear duct, as revealed by trajectory analysis of these OHCs (*Figure 6—figure supplement 5B,B'*). These E16, P1, and P7 OHCs were pooled with our own OHC-like cells, nascent HCs, SCs failing to become HCs, WT_P30 OHCs, and WT_P60 SCs to form a new dataset. Five main clusters were identified (*Figure 6F*): cluster 1 included 9/11 OHC-like cells, 11/16 nascent HCs, and the majority of wild-type OHCs at P1 (WT_P1 OHCs) (*Figure 6G*). Cluster 2 consisted of 2/11 OHC-like cells, 5/16 nascent HCs, 12/15 SCs failing to become HCs, and wild-type OHCs at E16 (WT_E16 OHCs). Majority of wild-type OHCs at P7 (WT_P7 OHCs) and WT_P30 OHCs formed cluster 3, suggesting that P7 OHCs were already well differentiated (*Jeng et al., 2020*). Cluster 4 contained the remaining fraction of WT_P1 OHCs and 3/15 SCs failing to become HCs, while majority of cells in cluster 5 were WT_P60 SCs. UMAP analysis suggested that OHC-like cells generally resembled WT_P1 OHCs the most. Further support for this came from the presence of *Insm1* mRNA and protein in OHC-like cells (*Figure 6— figure supplement 2A*, and *Figure 6—figure supplement 4A–B'''*); *Insm1* is a gene that is only transiently expressed in differentiating OHCs during late embryonic and perinatal ages (*Lorenzen et al., 2015*). We speculated that WT_P1 OHCs in cluster 1 originated from basal and middle turns as they more closely resembled WT_P7 and _P30 OHCs (*Figure 6*), whereas WT_P1 OHCs in cluster 4 that were grouped with the more immature E16 OHCs (*Figure 6F,G*) originated from apical turns.

Finally, trajectory analysis was applied to all the pooled cells (*Figure 6H,I*). Interestingly, two distinct tracks were observed: one reflected endogenous OHC development from E16 to P30 (black arrows in *Figure 6H*); the other tracked the reprogramming process from adult SCs into OHC lineage (dotted arrows in *Figure 6H*). Notably, in agreement with trajectory analysis described in *Figure 6—figure supplement 3*, OHC-like cells and nascent HCs were similar and intermingled with each other in both UMAP and trajectory analyses (*Figure 6G,H*).

## Neonatal-specific OHC genes are highly expressed in OHC-like cells

We next took a more focused look at the transcriptomic differences and similarities between OHC-like cells and WT_P1 OHCs. Based on two previous RNA-Seq studies (*Kolla et al., 2020*; *Wiwatpanit et al., 2018*), expression of 53 neonatal-specific OHC genes were further analyzed. These have been shown to be significantly enriched in WT_P0/P1 OHCs, but not in adult OHCs. The expression levels of these 53 genes in all 11 OHC-like cells was extracted from our raw data and expressed as transcripts per million (TPM). Those with TPM > 16 were defined as highly expressed genes; this threshold was chosen based on what gene expression level can typically be detected using RNA in situ hybridization with good signal to noise ratio in our hand. Our selection criteria for genes that were highly expressed in OHC-like cells thus were as follows: (1) that the gene's average TPM across all OHC-like cells was above 16; (2) that more than half of all OHC-like cells expressed this gene with TPM >16. If a gene did

not meet either of the criteria, then it was classed as a gene differently expressed between OHC-like cells and WT_P1 OHCs.

Twenty-six of 53 (49.1%) neonatal OHC genes, including *Insm1* (red arrow in *Figure 6—figure supplement 6A*), were found to be highly expressed in OHC-like cells; 21/53 (39.6%) of them, including *Bcl11b, Bmp2* and *Bmp7*, were expressed at low levels in these cells. Notably, *Bcl11b* is thought to be required for healthy stereocilia development (gray arrow in *Figure 6—figure supplement 6A*), as development of stereocilia in OHCs are defective in heterozygous *Bcl11b*± mice (*Okumura et al., 2011*). Only one of the two criteria were met in 6/53 (11.3%) genes (green arrows in *Figure 6—figure supplement 6A*). The numerical TPM values of these categories of neonatal OHC genes in OHC-like cells were included in *Supplementary file 5*. Together, based on the selected 53 OHC genes analyzed, the overall similarity between all the 11 OHC-like cells and neonatal OHCs is ~50 %.

As OHC-like cells only highly expressed about half of WT_P1 OHCs characteristic genes, we asked whether there was any overlap with neonatal IHC or vestibular HC-specific genes. Akin to before, based on three previous studies (*Burns et al., 2015*; *Kolla et al., 2020*; *Wiwatpanit et al., 2018*), 49 neonatal IHC and 26 utricle HC characteristic genes were further analyzed (*Figure 6—figure supplement 6B*). The same two criteria as above were applied to dictate whether a gene was expressed at a high or low level in OHC-like cells. None 0/49 (0%) of neonatal IHC-specific genes were expressed at a high level in OHC-like cells. Among the 26 neonatal utricle HC-specific genes, 4/26 (35.4%) were expressed at high levels in OHC-like cells, including *F730043M19Rik, Cfap161, Nme9*, and *1700001L19Rik* (red arrows in *Figure 6—figure supplement 6B*). As an internal reference, we also included nine genes, including *Slc26a5* and *Ocm*, that were highly expressed in OHC-like cells (marked by red line in *Figure 6—figure supplement 6B*). The numerical TPM values of these neonatal IHC and utricle HC-specific genes were included in *Supplementary file 6*. Taken together all the gene profiling results, we concluded that OHC-like cells resemble neonatal OHCs and not neonatal IHCs or utricle HCs.

## OHC-like cells are far less differentiated than adult wild-type OHCs

To complete our transcriptomic survey of OHC-like cells, we compared their expression patterns to mature P30_WT OHCs (*Figure 6—figure supplement 7A*). In total, 1998 genes were significantly higher in OHC-like cells than in P30_WT OHCs. Of note, 67 % (1339/1998) of these were overlapped with SC genes including *Fgfr3, Id1,* and *Id2* (purple arrows in *Figure 6—figure supplement 7A* and *Supplementary file 7*), most likely due to the persistent expression of these SC genes that were not able to be downregulated in the converted OHC-like cells. These remaining SC genes likely represented key barriers preventing OHC-like cells from undergoing further maturation.

Conversely, 361 genes were expressed at a significantly lower level in OHC-like cells as compared to WT_P30 OHCs. Notably, 49.6 % (179/361) of these were overlapped with OHC-enriched genes, including *Slc26a5* (Prestin), *Kcna10,* and *Lmo7* (green arrows in *Figure 6—figure supplement 7A*). Relevantly, *Lmo7* mutant mice have been reported to show abnormalities in HC stereocilia (*Du et al., 2019*) and *Kcna10* mutants exhibit mild auditory dysfunction (*Lee et al., 2013*). Thus, the relatively low expression of these functional proteins might account in part for the immature status of the new OHC-like cells. *Supplementary file 7* contained the complete list of genes differently expressed genes between OHC-like cells and WT_P30 OHCs. GO analysis results showed that genes expressed at higher levels in OHC-like cells were associated with cell adhesion (*Figure 6—figure supplement 7B*). The complete gene list associated with the GO cell adhesion category could be found in *Supplementary file 8*.

To conclude, the current differentiation or cell fate conversion status of the OHC-like cells was simply summarized in *Figure 7*, more manipulations are still needed to further promote maturation of OHC-like cells. Decreasing the remaining SC genes and/or increasing OHC genes represent actionable and promising avenues.

## Discussion

In summary, the merit of our study lies in the demonstration of effective conversion of adult cochlear SCs (mainly PCs and DCs) into Prestin+ OHC -like cells after OHC damage in vivo. This experimental

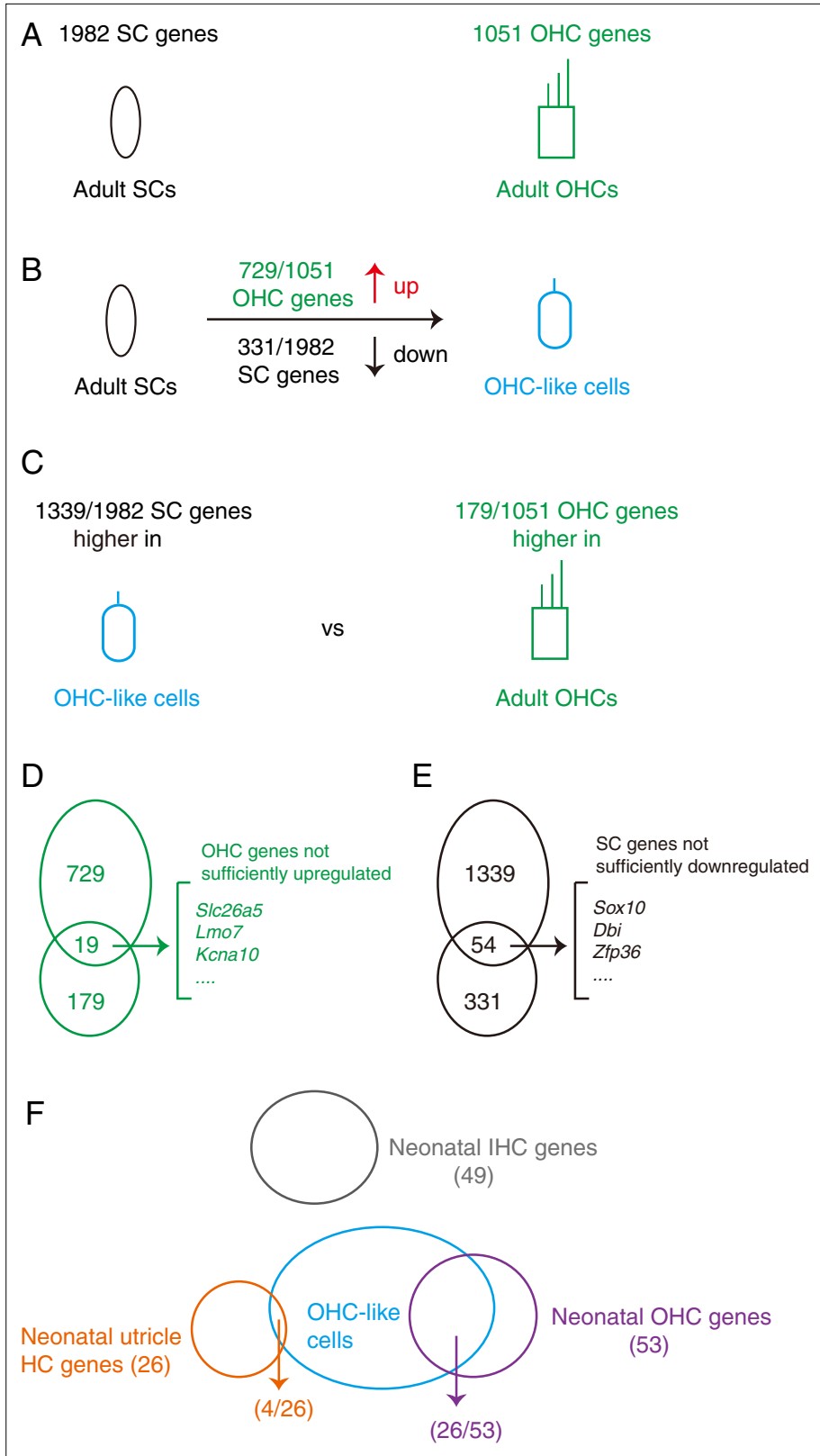

**Figure 7.** Summary of the differentiation status of the OHC-like cells. (**A**) Comparison of transcriptomic profiles revealed 1982 SC and 1,051 OHC differentially expressed genes. (**B**) Compared to adult SCs, OHC-like cells upregulated 729 OHC genes and downregulated 331 SC genes, thus having become more OHC-like and less SC like. (**C**) Compared to adult OHCs, OHC-like cells expressed 1339 SC genes at higher levels, and 179 OHC genes

*Figure 7 continued on next page*

*Figure 7 continued*

at lower levels, showing that OHC status had not yet been achieved. (**D**) 19 OHC genes intersected between the upregulated 729 OHC genes (as compared to adult SCs) and the under-expressed 179 OHC genes (as compared to adult OHCs), suggesting that although these genes were up-regulated in OHC-like cells, their expression levels were still lower than that expected for adult OHCs. (**E**) Similar to (**D**), 54 SC genes overlapped between 1339 and 331 SC genes that were found to be at higher levels than expected for adult OHCs, but at lower levels than expected for adult SCs. These SC genes had thus successfully been downregulated, but not sufficiently so to reach the low levels in adult OHCs. (**F**) OHC-like cells expressed 26 of the 53 neonatal OHC genes, but only expressed 4 of the 26 neonatal utricle HC genes and did not express any neonatal IHC genes.

paradigm is clinically relevant and suggests that *Atoh1* and *Ikzf2* are candidate therapeutics for OHC regeneration.

## Potential roles of Atoh1 and Ikzf2 in cell-fate transition from adult cochlear SCs into OHC-like cells

During normal HC development, Atoh1 is expressed for approximately 1 week, beginning at the basal end of the cochlea and progressively extending to the apex; expression is then down-regulated following the same temporal basal-to-apical sequence. During this period, Atoh1 is reported to be involved in at least three age-dependent processes: (1) specifying general HC fate, (2) maintaining the survival of HC progenitors (short term) and OHCs (long term), and (3) organizing the formation of hair bundles (*Bermingham et al., 1999*; *Cai et al., 2013*; *Woods et al., 2004*). While *Atoh1* is sufficient to convert neonatal PCs and DCs into HCs (*Liu et al., 2012a*), it is insufficient to do so in adults. Ikzf2 alone also cannot convert adult SCs into HCs but the combination both *Atoh1* and *Ikzf2* is synergistic and can do so in adults. Whether these effects are similar to the synergistic effects between Atoh1 and Pou4f3, which were recently reported during normal cochlear HC development (*Yu et al., 2021*), will require further investigation.

Ikzf2 is necessary for OHC maturation, and an Ikzf2 point-mutation model, Ikzf2 [cello/cello], displays early-onset hearing loss and diminished Prestin expression; dual to this, ectopic Ikzf2 induces Prestin expression in IHCs (*Chessum et al., 2018*). Whether Prestin is a direct target of Ikzf2 is unclear, but the onset of Ikzf2 expression at neonatal ages and its specific and permanent expression in OHCs suggest that Ikzf2 is a key TF required to specify and maintain the OHC fate and/or repress the IHC fate. Therefore, we speculate that Ikzf2 was responsible for directing nascent HCs into the OHC differentiation track. In accord with this, Ikzf2 alone was able to induce Prestin expression in wild-type IHCs but not adult SCs, likely due to that IHCs are already in the general 'HC differentiation track' and thus more are poised to express OHC genes. In contrast to the ability of Atoh1, which takes approximately three weeks to convert SC to HC fate in the neonatal cochlea (*Liu et al., 2012a*), the combined effects of Atoh1 and Ikzf2 in mediating this cell-fate switch in the adult cochlea occurred within a shorter time window (approximately 12 days). This could be a result of the synergistic actions between Atoh1 and Ikzf2.

## Active OHC death promotes transformation from adult SCs into OHC-like cells

In our study, DT treatment led to selective OHC death, which was defined as active OHC death. In opposition to this, OHC death occurred as a side effect of overexpressing Ikzf2 alone or Ikzf2 and Atoh1 concurrently (*Figure 2*), which we refer to as passive OHC death. While the detailed mechanisms are unclear, passive OHC death has been observed in multiple instances upon: (1) overexpressing Atoh1 in neonatal or juvenile cochlear SCs (*Liu et al., 2012a*), (2) triggered neonatal SC proliferation (*Liu et al., 2012b*), and (3) conditional SC damage at postnatal ages (*Mellado Lagarde et al., 2013*; *Mellado Lagarde et al., 2014*). Our data showed that only OHC death that was 'active' could significantly increase numbers of OHC-like cells upon Atoh1 and Ikzf2 overexpression in adult SCs (*Figure 4I*).

How to interpret the differential effects of active and passive OHC death? It is known that DT, but not neomycin, induced HC death can trigger Wnt-dependent HC regeneration from SCs in neonatal mice (*Hu et al., 2016*). This exemplifies that the precise way that HC dies can drastically affect how cochlear SCs respond to HC death. DT treatment causes cell membrane rupture leading to the release

of cellular components in the extracellular space. We speculated that SCs overexpressing Atoh1 and Ikzf2 responded favorably to yet unidentified factors released from OHCs during active cell death. Such factors may not be made accessible to SCs upon passive OHC death.

We also conjectured that the timing of OHC active death in relation to the SC cellular status was likely the key. When active OHC death occurred by DT injection, Atoh1 and Ikzf2 had already been overexpressed in adult SCs for 6 days (*Figure 4B*). This particular temporal coupling may have permitted those adult SCs to respond more favorably to the unidentified factors released from nearby dying OHCs, to ultimately become OHC-like cells. When no active death was induced, and instead OHCs died passively due to secondary effect of ectopic Atoh1 and Ikzf2 expression, it may have been the timing between SC gene induction and OHC death that was unfavorable. Future studies will be needed to determine the main contributing factors dictating conversion efficiency or whether these hypothesis have any ground at all.

## Degree of cell fate conversion of the OHC-like cells

By comparing the transcriptomic profiles between adult endogenous OHCs and SCs, we identified 1051 OHC and 1982 SC genes, respectively (*Figure 6—figure supplement 1A* and *Supplementary file 1*). We then used the degree of OHC gene upregulation and SC gene downregulation to estimate the extent of cell fate conversion from adult SCs to OHC-like cells (*Figure 7A*). Promisingly, relative to adult SCs (primarily PCs and DCs), 69.4 % (729/1051) of OHC genes were significantly up-regulated in OHC-like cells (*Figure 7B*). A more limited number of SC genes, 16.7 % (331/1982), were significantly down-regulated in OHC-like cells (*Figure 7B*). It appears that our reprogramming manipulations were more efficient at endowing OHC-like status to SCs, than 'erasing/making them forget' their SC status.

To compare the transcriptomic profiles between OHC-like cells and adult endogenous OHCs, we selected endogenous OHCs at P30, but not P60, because OHC-like cells were derived from SCs at P30 and thus their intrinsic age might be ~P30, although the mice were analyzed at P60. Nevertheless, cochlear development is complete by P30, and wild-type OHCs at P30 and P60 are expected to differ minimally. Consistently, it is reported that transcriptomic profiles of adult cochlear SCs at P60 and P120 are indistinguishable from each other (*Hoa et al., 2020*). Thus, compared to adult endogenous OHCs, 17 % (179/1051) of OHC genes were expressed at lower levels in OHC-like cells (*Figure 7C*). A substantial proportion of SC genes 67.6 % (1339/1982) remained expressed at much higher level in OHC-like cells than that in adult endogenous OHCs (*Figure 7C*). In addition, 19 OHC genes, including *Slc26a5, Lmo7,* and *Kcna10*, were not sufficiently up-regulated in OHC-like cells (*Figure 7D*). Notably, Ocm was not included in the 19 OHC genes, suggesting that Ocm was better upregulated than *Slc26a5* in the OHC-like cells. Likewise, 54 SC genes, including *Sox10, Dbi,* and *Zfp36*, were not sufficiently downregulated in OHC-like cells (*Figure 7E*).

In logic accord to only finding moderate transcriptomic similarities between OHC-like cells and adult endogenous OHCs, we identified stronger similarities with neonatal OHCs via different approaches (*Figure 7F*). Firstly, our integrated UMAP analysis embedded OHC-like cells closest to WT_P1 OHCs. Secondly, we found that OHC-like cells had upregulated 26 of the 53 known characteristic neonatal OHC genes. For instance, *Insm1* was upregulated but not *Bcl11b* (*Figure 6—figure supplement 6A*). This was consistent with that *Insm1* is turned on earlier than *Bcl11b* during OHC development (*Wiwatpanit et al., 2018*).

We also investigated how similar OHC-like cells were to neonatal IHCs or utricle HCs. To do this, we first identified genes specifically enriched in neonatal IHCs or utricle HCs (*Figure 6—figure supplement 6B*) and extracted their expression level in OHC-like cells. We found that neonatal IHC and utricle HC-specific genes were minimally expressed, if at all, in OHC-like cells. In addition, we noted that *Cfap161*, one of the four neonatal utricle HC genes expressed in OHC-like cells, is recently reported to be expressed in many cells with cilia structure including cochlear HCs via immunostaining assay (*Beckers et al., 2021*). Such discrepancies may be due to difference in the sensitivity of different techniques used. In addition, the current HC subtype specific genes were selected from three studies, but it is always challengeable and has limitations to compare data from different studies, especially the data are generated by different sequencing methods (*Kolla et al., 2020*). Nevertheless, more detailed gene profiling studies of different inner ear HCs at various developmental ages are needed to more precisely define some markers of individual HC subtypes.

Despite many OHC genes were significantly upregulated, numerous SC genes were retained in the OHC-like cells. Then, how best to define the cellular status of OHC-like cells? Alternatively, are they more akin to OHCs or new hybrid OHCs/SCs taking a novel differentiation path? In our mind, there is no consensus yet to these questions, but great scope for exploration and discovery. We postulate that transcriptomic assays alone will likely be insufficient to define what constitutes a cell type for our purposes. Macro properties such as anatomical features, especially hair bundle structure in apical surface of HCs, as well as electrophysiological properties, will need to be measured. With that said, the most indisputable proof of an OHC cell state should be hearing recovery after OHC damage. In summary, our data supported that OHC-like cells resemble neonatal OHCs and not neonatal IHCs or utricle HCs.

### Potential future approaches to promote more complete differentiation of OHC-like cells

How to further promote differentiation of these OHC-like cells? This is the key question as the ultimate goal is to convert adult cochlear SCs into functional OHCs. To begin with, *Atoh1* is only transiently expressed in cochlear OHCs during normal development (*Cai et al., 2013*; *Driver et al., 2013*; *Matei et al., 2005*). However, our model induced permanent Atoh1 expression, potentially hindering maturation of OHC-like cells. Therefore, transient Atoh1 expression combined with permanent Ikzf2 expression may be the focus of future work.

In conjunction, we speculated that combining other candidate genes, together with *Atoh1* and *Ikzf2*, would likely drive OHC-like cells to further mature. Promising candidate genes could include ones necessary for hair bundle organization and growth, as our results highlighted the reduced number and deformity of the stereocilia in OHC-like cells (*Figure 5*). For instance, the neonatal OHC gene *Bcl11b* was missing in OHC-like cells (gray arrows in *Figure 6—figure supplement 6A*) and heterozygous *Bcl11b*± mice exhibit impaired formation of hair bundles in OHCs (*Okumura et al., 2011*). Simultaneous expression *Atoh1*, *Ikzf2*, and *Bcl11b* might therefore have higher reprogramming efficiency than Atoh1 and Ikzf2. Encouragingly, OHC-like cells did express multiple MET-channel-related proteins such as *Cib2* and *Tmc1* (*Jia et al., 2020*; *Li et al., 2019*; *Pan et al., 2013*). This aligned with the previous report that MET-channel assembly does not require normal hair bundle morphology (*Cai et al., 2013*).

As hair bundles mostly develop after birth, we also hypothesized that genes crucial for hair bundle organization are likely to be differentially expressed between P1 and P14/P30 OHCs. In terms of phenotypic assessment of functional hair bundles, rapid loss-of-function test will be performed following the protocols described in our previous pilot studies (*Wang et al., 2021*; *Zhang et al., 2018*). The phenotypes we hope to focus on will include shorter or disorganized OHC hair bundles, and ones with reduced numbers of OHC hair bundles at P14 or P30. Finally, we took note of the many SC genes expressed in OHC-like cells, including *Fgfr3*, *Id1*, and *Id2* (*Supplementary file 7*). Fgf signaling is known to be involved in SC fate maintenance (*Doetzlhofer et al., 2009*; *Jacques et al., 2007*; *Shim et al., 2005*). Id gene family is known to repress HC formation by negatively regulating *Atoh1* (*Jones et al., 2006*). Decreasing Fgfr3 and Id1/2 may promote further maturation of OHC-like cells.

In summary, our current study provided a genetic model for OHC damage and subsequent regeneration in adult mouse cochlea. Building on this model and adding further genetic manipulations represent a promising avenue to regenerate functional OHCs.

## Materials and methods

**Key resources table**

| Reagent type (species) or resource | Designation | Source or reference | Identifiers | Additional information |
|---|---|---|---|---|
| Genetic reagent (*Mus. musculus*) | FVB-Tg(Atoh1-cre/ERT)1Sbk/Mmnc | MMRRC | Stock#:029581-UNC | RRID:MMRRC_029581-UNC |
| Genetic reagent (*Mus. musculus*) | Tg(Fgfr3-icre/ERT2)4-2Wdr | Jackson Lab | Stock#:025809 | RRID:IMSR_JAX:025809 |

*Continued on next page*

*Continued*

| Reagent type (species) or resource | Designation | Source or reference | Identifiers | Additional information |
|---|---|---|---|---|
| Genetic reagent (*Mus. musculus*) | Ai9 | Jackson Lab | Stock#:007909 | RRID:IMSR_JAX:007909 |
| Genetic reagent (*Mus. musculus*) | *Slc26a5*$^{CreER/+}$ | *Fang et al., 2012* | PMID:21954035 |
| Genetic reagent (*Mus. musculus*) | CAG-LSL-*Atoh1+* | *Liu et al., 2012a* | PMID:22573682 |
| Genetic reagent (*Mus. musculus*) | *Rosa26*$^{CAG-LSL-Ikzf2/+}$ | Jackson Lab | Stock#:036272 | Knock-in mouse line that will be available soon |
| Genetic reagent (*Mus. musculus*) | *Slc26a5*$^{DTR/+}$ | Jackson Lab | Stock#:036271 | Knock-in mouse line that will be available soon |
| Antibody | anti-HA (Rat monoclonal) | Roche | Cat.#:11867423001 | RRID:AB_390918 |
| Antibody | anti-Prestin (Goat polyclonal) | Santa Cruz | Cat.#:sc-22692 | RRID:AB_2302038 |
| Antibody | anti-Myosin-VI (Rabbit polyclonal) | Proteus Bioscience | Cat.#:25–6791 | RRID:AB_2314836 |
| Antibody | anti-Myosin-VIIa (Rabbit polyclonal) | Proteus Bioscience | Cat.#:25–6790 | RRID:AB_10015251 |
| Antibody | anti-Insm1 (Guinea pig) | A kind gift from Dr. Shiqi Jia and Dr. Carmen Birchmeier | *Jia et al., 2015* | PMID:25828096 |
| Antibody | anti-Parvalbumin (Mouse monoclonal) | Sigma-Aldrich | Cat.#:P3088 | RRID:AB_477329 |
| Antibody | anti-Rbm24 (Rabbit polyclonal) | Proteintech | Cat.#:18178–1-AP | RRID:AB_2878513 |
| Antibody | anti-Calbindin (Rabbit monoclonal) | Sigma-Aldrich | Cat.#:C9848 | RRID:AB_2314067 |
| Antibody | anti-vGlut3 (Rabbit polyclonal) | Synaptic System | Cat.#:135,203 | RRID:AB_887886 |
| Antibody | anti-Ocm (Rabbit) | Swant | Cat.#:OMG4 | RRID:AB_10000346 |
| Antibody | anti-Slc7a14 (Rabbit polyclonal) | Sigma-Aldrich | Cat.#:HPA045929 | RRID:AB_2679501 |
| Antibody | anti-Ctbp2 (Mouse monoclonal) | BD Biosciences | Cat.#:612,044 | RRID:AB_399431 |
| Commercial assay, kit | SMART-Seq HT Kit | Takara | Cat.#:634,437 | For reverse-transcription and cDNA amplification |
| Commercial assay, kit | TruePrep DNA Library Prep Kit V2 for Illumina | Vazyme | Cat.#:TD503-02 | For sequencing library construction |
| Commercial assay, kit | TruePrep Index Kit V2 for Illumina | Vazyme | Cat.#:TD202 | For sequencing library construction (index) |

## Generation of *Rosa26*$^{CAG-LSL-Ikzf2/+}$ and *Slc26a5*$^{DTR/+}$ knock-in mouse strains using CRISPR/Cas9

The *Rosa26*-CAG-Loxp-stop-Loxp-*Ikzf2*\*3xHA-T2A-Tdtomato/+ (*Rosa26*$^{CAG-LSL-Ikzf2/+}$) knock-in mouse strain was produced by co-injecting one sgRNA against the *Rosa26* locus (5'-actccagtctttctagaaga-3'), *donor DNA* (*Figure 1—figure supplement 1*), and *Cas9* mRNA into one-cell-stage mouse zygotes. A similar approach was followed to generate the *Slc26a5*-P2A-DTR/+ (*Slc26a5*$^{DTR/+}$) knock-in mouse line. In this case, sgRNA targeting the *Slc26a5* (Prestin) locus was used: 5'-CGAGGCATAAAGGCCCT-GTA-3'. Information regarding the donor DNA is described in *Figure 3—figure supplement 1*. F0 mice of both strains were screened for potential successful insertion by performing junction PCR; this was followed by crossing the positive F0 mice with wild-type C57BL/6 mice to assess the germ-line status and to produce stable F1 mice. F1 mice was further confirmed via junction PCR. Moreover, lack of random insertion of donor DNA in the genome of these F1 mice was confirmed using Southern blotting (*Figure 1—figure supplement 1D,E* and *Figure 3—figure supplement 1D,E*), which was performed according to our previously published protocol (*Li et al., 2018b*). Both mouse strains were PCR-genotyped using tail DNA (representative gel images of PCR products were presented in *Figure 1—figure supplement 1F* and *Figure 3—figure supplement 1F*). Details regarding primer

sequences can be found in *Supplementary file 9*. All mice were bred and raised in SPF-level animal rooms, and animal procedures were performed according to the guidelines (NA-032–2019) of the IACUC of Institute of Neuroscience (ION), CAS Center for Excellence in Brain Science and Intelligence Technology, Chinese Academy of Sciences.

## Sample processing, histology, and immunofluorescence assays, and cell counting

Adult mice were anesthetized and sacrificed during which hearts were perfused with 1× phosphate-buffered saline (PBS) and fresh 4 % paraformaldehyde (PFA) to completely remove blood from the inner ear. Inner ear tissues were then carefully dissected out, re-fixed with fresh 4 % PFA overnight at 4 °C, and washed thrice with 1× PBS. The tissues were decalcified first with 120 mM EDTA for 2 days at 4 °C until they were soft enough for micro-dissection and whole-mount preparation. Immunostaining was then performed with the following primary antibodies: anti-HA (rat, 1:200, 11867423001, Roche, RRID:AB_390918), anti-Prestin (goat, 1:1000, sc-22692, Santa Cruz, RRID:AB_2302038), anti-Myo6 (rabbit, 1:500, 25–6791, Proteus Bioscience, RRID:AB_2314836), anti-Myo7a (rabbit, 1:500, 25–6790, Proteus Bioscience, RRID:AB_10015251), anti-Ocm (rabbit, 1:500, OMG-4, Swant, RRID:AB_10000346), anti-Insm1 (guinea pig, 1:6000; a kind gift from Dr. Shiqi Jia from Jinan University, Guangzhou, China, and Dr. Carmen Birchmeier from Max Delbrück Center for Molecular Medicine, Berlin, Germany) (*Jia et al., 2015*), anti-Ctbp2 (mouse, 1:500, 612044, BD Bioscience, RRID:AB_399431), anti-Parvalbumin (mouse, 1:500, P3088, Sigma-Aldrich, RRID:AB_477329), anti-Rbm24 (rabbit, 1:500, 18178–1-AP, Proteintech, RRID:AB_2878513), anti-Slc7a14 (rabbit, 1:500, HPA045929, Sigma-Aldrich, RRID:AB_2679501), anti-Calbindin (rabbit, 1:500, C9848, Sigma-Aldrich, RRID:AB_2314067), and anti-vGlut3 (rabbit, 1:500, 135203, Synaptic System, RRID:AB_887886). Cochlear tissues were counterstained with Hoechst 33,342 solution (1:1000, 62249, Thermo Scientific, RRID:AB_2651135) to visualize nuclei, and were mounted with Prolong gold antifade medium (P36930, Thermo Scientific, RRID: SCR_015961). Nikon C2, TiE-A1, and NiE-A1 plus confocal microscopes were used to capture images.

Each whole-mount preparation of individual cochlear duct was divided into three parts and imaged at 10× magnification using a confocal microscope. In each acquired image, a line was drawn passing through the center of the IHCs and OHCs to measure the full length of each cochlear duct. They were then divided into basal, middle, and apical portions of equal length. To assess the extent of OHC death in the *Slc26a5$^{DTR/+}$* model (*Figure 3*), the ratio of the OHC to IHC numbers was used as a proxy as IHC were not affected by DT treatment. To this end, OHC and IHC numbers within a scanning region (60×, confocal microscopy) were computed. This was repeated for two areas in each cochlear turn and mean number was calculated. To count Prestin+ IHCs in Atoh1-CreER+; *Rosa26$^{CAG-LSL-Ikzf2/+}$* mice (*Figure 1D*), and Tdtomato+/HA+ cells, OHC-like cells, and nascent HCs in Fgfr3-Atoh1-Ikzf2 and Fgfr3-Atoh1-Ikzf2-DTR mice (*Figures 2 and 4*), the entire cochlear duct of each mouse was scanned to minimize variation between different replicates (60 ×, confocal microscope) and cells were counted manually. All cell counting data were presented as means ± SEM. Statistical analyses were performed using one-way ANOVA, followed by a Student's t-test with Bonferroni correction. All statical analyses were implemented using GraphPad Prism 6.0.

## Single-cell RNA sequencing and bioinformatic analysis

Single-cell RNA sequencing was performed in three different mouse lines; each time the cells of interest were identifiable by Tdtomato expression. Cells originated: (1) from P30 *Slc26a5$^{CreER/+}$*; Ai9/+ mice that were injected with tamoxifen at P20 and P21; in this case, Tdtomato+ cells were exclusively OHCs due to selective Cre expression in OHCs (*Fang et al., 2012*). (2) from P60 Fgfr3-iCreER+; Ai9/+ mice that were administered tamoxifen at P30 and P31; in this instance, Tdtomato+ cells within the cochlear sensory epithelium were SCs (primarily PCs and DCs) according to our previous reports (*Liu et al., 2012a*; *Liu et al., 2012b*). (3) from P60 Fgfr3-Atoh1-Ikzf2-DTR mice that were administered tamoxifen at P30 and P31 and then DT at P36. In this case, Tdtomato+ cells within the cochlear sensory epithelium included OHC-like cells, nascent HCs, and SCs failing to become HCs.

The cochlear sensory epithelium from each mouse was dissected out, digested, and single-cell suspensions were prepared according to our previously described protocol (*Li et al., 2020b*). All the aforementioned Tdtomato+ cells were picked under a fluorescence microscope (M205FA, Leica) as

illustraed in *Figure 6A*. We picked 17 wild-type adult OHCs at P30, 16 wild-type adult SCs at P60, and 42 Tdtomato+ cells from the Fgfr3-Atoh1-Ikzf2-DTR model at P60 (*Figure 6D*). Once the cells were picked, they were immediately subjected to reverse-transcription and cDNA amplification using a Smart-Seq HT kit (Cat# 634437, Takara). The cDNAs (1 ng each) were tagmented using a TruePrep DNA Library Prep Kit V2 for Illumina (Cat# TD503-02, Vazyme) and a TruePrep Index Kit V2 for Illumina (Cat# TD202, Vazyme). The final libraries were subject to paired-end sequencing on the Illumina Novaseq platform. Each library was sequenced leading to 4 G of raw data.

FastQC (v0.11.9) and trimmomatic (v0.39) were used for quality control of raw sequencing data. High-quality mapping to the mouse reference genome (GRCm38) was achieved for 70–80% of the reads using the Hisat2 (v2.1.0) program with default parameters. Raw counts were calculated using HTSeq (v0.10.0), and gene-expression levels were estimated by using StringTie (v1.3.5) with default parameters. Gene abundances were presented as transcript per million (TPM) values. Differentially expressed genes (DEGs) were analyzed using R package 'DESeq2' (p<0.001, absolute value of (log2 fold change) > 2). Based on the identified DEGs, biological process enrichment was evaluated (p<0.001, adjusted using FDR correction) by using DAVID (Database for Annotation, Visualization and Integrated Discovery). Extensive quality checks of the data were carried out throughout the analyses. Detailed information regarding mapping rate to mouse genome, neumbers of detected genes with TPM > 1 and TPM > 10, as well as cell names and their corresponding cell identities could be found in *Supplementary file 10*. The raw data generated from this single-cell RNA sequencing study is accessibly in the GEO (Gene Expression Omnibus) database under Accession No. GSE161156.

To analyze the previously published RNA-seq data of wild-type OHCs at E16, P1, and P7 in a recent study (*Kolla et al., 2020*), the Seurat processing pipeline (R package v3.0) was applied. To adequately compare the transcriptomic profiles obtained from 10× genomics and smartseq methods, we first integrated them by using the Seurat functions 'SCTransform', 'FindIntegrationAnchors' (k.filter = 30), and 'IntegrateData'. Principal component analysis (PCA) was performed by executing the 'RunPCA' function, and the top 20 principal components (PCs) were used for further dimensionality reduction using UMAP (via 'RunUMAP'). Unsupervised clustering was performed using the 'FindNeighbors' and 'FindClusters' function (resolution = 0.5 for *Figure 6—figure supplement 5A*, 0.8 for *Figure 6F*, or 1.5 for *Figure 6D*).

Furthermore, E16_WT OHCs were defined as cells in which the expression level of *Insm1*, *Myo6*, and *Atoh1* were all above zero; P1_WT OHCs were identified as those with above zero expression of *Bcl11b*, *Myo6*, *Myo7a*, and *Atoh1*. Finally, cells expressing above zero level of *Slc26a5* (Prestin), *Myo6*, *Ocm*, and *Ikzf2* were classified as P7_WT OHCs. Trajectory analysis was performed using Monocle (R package v2.0). Pre-processed Seurat datasets were imported into Monocle. The top 2000 most variable genes identified from the Seurat object were used as data input to Monocle. Cells were ordered along a pseudotime axis by using the 'orderCells' function in Monocle. The pseudotime scale in monocle is an arbitrary unit, and it is simulated according to the developmental order (or age/differentiation status) of each cell.

## ABR measurement

ABR were measured in response to the following sound frequencies: 4 k, 5.6 k, 8 k, 11.3 k, 16 k, 22.6 k, and 32 k Hz, following our previously published (*Li et al., 2018b*). To assess statistical significance between responses of the same frequency between conditions, Student's t-tests were performed (*Figures 3C and 5D*).

## Tamoxifen and DT treatment

Tamoxifen (Cat# T5648, Sigma-Aldrich) was dissolved in corn oil (Cat# C8267, Sigma-Aldrich) and injected intraperitoneally at 3 mg/40 g body weight (for P0 and P1 mice) or at 9 mg/40 g body weight (for P20 and P21, and P30 and P31 mice). DT (Cat# D0564, Sigma-Aldrich) dissolved in 0.9 % NaCl solution was also delivered through intraperitoneal injection, at a dose of 20 ng/g body weight. The mice were sacrificed at P42, P46, or P60.

## SEM data collection and analysis

SEM was performed following the protocol reported previously (*Parker et al., 2016*). Briefly, we made holes at the cochlear apex. Cochlear samples were then washed gently with 0.9 % NaCl

(Cat#10019318, Sinopharm Chemical Reagent Co, Ltd.) and fixed overnight with 2.5 % glutaraldehyde (Cat# G5882, Sigma-Aldrich) at 4 °C. On the following day, the cochlear samples were washed thrice with 1× PBS and subsequently decalcified using 10 % EDTA (Cat# ST066, Beyotime) for 1 day, then refixed for 1 h with 1 % osmium tetroxide (Cat#18451, Ted Pella), and subject to a second fixation with thiocarbohydrazide (Cat#88535, Sigma-Aldrich) for 30 min and further fixed fixation for 1 hr with osmium tetroxide. Next, the cochlear samples were dehydrated using a graded ethanol series (30%, 50%, 75%, 80%, 95 %; Cat#10009259, Sinopharm Chemical Reagent Co, Ltd.) at 4 °C with each incubation step lasting 30 min. The series was completed by fully dehydrating the samples in 100 % ethanol (three times, 30 min for each) at 4 °C. The cochlear samples were then dried in a critical point dryer (Model: EM CPD300, Leica), after which whole-mount cochlear samples were prepared under a microscope to ensure that hair bundles were facing upward. Finally, samples were treated with a turbomolecular pumped coater (Model: Q150T ES, Quorum). The prepared samples were then scanned using a field-emission SEM instrument (Model: GeminiSEM 300, ZEISS).

## Acknowledgements

We thank Drs. Qian Hu, Yu Kong, and Xu Wang from Optical Imaging and EM Facility of the Institute of Neuroscience (ION) for support with the image analysis; Dr. Hui Yang (Principal Investigator at the ION) for sharing the zygote microinjection system used to generate the knockin mice; Ms. Qian Liu (from the Department of Embryology of the ION animal center) for helping us in transplanting zygotes into pseudopregnant female mice; Dr. Shiqi Jia (Jinan University, Guangzhou, China) and Dr. Carmen Birchmeier (Max Delbrück Center for Molecular Medicine, Berlin, Germany) for kindly providing the anti-Insm1 antibody; and Ms.Virginia M S Rutten (HHMI-Janelia Research Campus, Ashburn, VA, USA) for assistance in editing the text.

## Additional information

### Competing interests

Suhong Sun, Shuting Li, Zhengnan Luo, Minhui Ren, Shunji He, Guangqin Wang, Zhiyong Liu: We filed an auditory hair cell regeneration patent based on the key findings of this manuscript..

### Funding

| Funder | Grant reference number | Author |
|---|---|---|
| National Natural Science Foundation of China | 81771012 | Zhiyong Liu |
| Ministry of Science and Technology of the People's Republic of China | 2017YFA0103901 | Zhiyong Liu |
| Chinese Academy of Sciences | XDB32060100 | Zhiyong Liu |
| Shanghai Municipal Bureau of Quality and Technical Supervision | 2018SHZDZX05 | Zhiyong Liu |
| Shanghai Jiao Tong University | SSMU-ZLCX20180601 | Zhiyong Liu |
| Boehringer Ingelheim | DE811138149 | Zhiyong Liu |

The funders had no role in study design, data collection and interpretation, or the decision to submit the work for publication.

### Author contributions

Suhong Sun, Zhengnan Luo, Data curation, Formal analysis, Writing - review and editing; Shuting Li, Data curation, Formal analysis, Investigation, Writing - review and editing; Minhui Ren, Formal analysis, Software; Shunji He, Guangqin Wang, Formal analysis, Methodology; Zhiyong Liu, Conceptualization,

Formal analysis, Funding acquisition, Investigation, Project administration, Supervision, Writing - original draft, Writing - review and editing

## Author ORCIDs
Suhong Sun http://orcid.org/0000-0003-0059-4534
Shuting Li http://orcid.org/0000-0003-3438-1588
Zhengnan Luo http://orcid.org/0000-0002-8204-6277
Zhiyong Liu http://orcid.org/0000-0002-9675-1233

## Ethics
All mice were bred and raised in SPF level animal rooms and animal procedures were performed according to guidelines (NA-032-2019) of the IACUC of Institute of Neuroscience (ION), Chinese Academy of Sciences.

## Decision letter and Author response
Decision letter https://doi.org/10.7554/eLife.66547.sa1
Author response https://doi.org/10.7554/eLife.66547.sa2

---

# Additional files

## Supplementary files
• Supplementary file 1. List of all genes that were expressed at higher ($P < 0.001$) levels in WT_P30 OHCs as compared to WT_P60 SCs, and vice versa. The most differentially expressed genes were listed in *Figure 6—figure supplement 1A*.

• Supplementary file 2. GO analysis of genes that were significantly enriched in WT_P30 OHCs, relative to WT_P60 SCs. Top GO terms were shown in *Figure 6—figure supplement 1B*.

• Supplementary file 3. GO analysis of genes that were significantly enriched in OHC-like cells, relative to WT_P60 SCs. Top GO terms were presented in *Figure 6—figure supplement 2B*.

• Supplementary file 4. List of genes expressed at higher level in OHC-like cells relative to WT_P60 SCs, and vice versa. In addition, these genes which were overlapped with OHC or SC-enriched genes (*Figure 6—figure supplement 2A*) are also included in a separate tab in the file.

• Supplementary file 5. TPM values in OHC-like cells of 53 neonatal OHC genes, as presented in *Figure 6—figure supplement 6A*.

• Supplementary file 6. TPM values in OHC-like cells of 9 pan-HC or OHC-enriched genes, 49 neonatal IHC genes, and 26 neonatal utricle HC genes, as presented in *Figure 6—figure supplement 6B*.

• Supplementary file 7. List of all genes that were differently expressed between OHC-like cells and WT_P30 OHCs. In addition, we also included those differently expressed genes that were overlapped with SC or OHC genes (presented in *Figure 6—figure supplement 7A*), as well as the 19 OHC genes (not sufficiently upregulated) or 54 SC genes (not sufficiently downregulated) in OHC-like cells.

• Supplementary file 8. GO analysis of genes that were significantly enriched in OHC-like cells relative to WT_P30 OHCs. The results were presented in *Figure 6—figure supplement 7B*.

• Supplementary file 9. Genotyping primers, and sizes of PCR amplicons for the various knock-in and transgenic mouse strains used in this study.

• Supplementary file 10. Detailed information of all the 75 single cells that were subject to smartseq analysis. Notably, the OHC-like cells, nascent HCs and SCs failing to become HCs were picked from the same model and labelled exp-1 to exp-42 in the file. Their cell identity was defined after bioinformatic analysis.

• Transparent reporting form

## Data availability
Sequencing data have been deposited in GEO under accession codes: GSE161156.

The following previously published datasets were used:

| Author(s) | Year | Dataset title | Dataset URL | Database and Identifier |
|---|---|---|---|---|
| Kolla | 2020 | Characterization of cochlear development at the single cell level | https://www.ncbi.nlm.nih.gov/geo/query/acc.cgi?acc=GSE137299 | NCBI Gene Expression Omnibus, GSE137299 |

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
