## [Decision Letter]

**Acceptance summary:**

The transcription factor, Atoh1, is capable of regenerating hair cells in neonatal but not mature mammalian cochleae after hair cell loss. Here, Sun et al. demonstrated that a combination of Atoh1 and a transcription factor important for specifying outer hair cell (OHC) fate, Ikzf2, can convert supporting cells in damaged adult cochleae to form hair cells that exhibit OHC characteristics. This experimental approach is a positive step towards the goal of alleviating hearing loss.

**Decision letter after peer review:**

Thank you for submitting your article "Dual expression of Atoh1 and Ikzf2 promotes transformation of adult cochlear supporting cells into outer hair cells" for consideration by *eLife*. Your article has been reviewed by 3 peer reviewers, including Doris Wu as the Reviewing Editor and Reviewer #1, and the evaluation has been overseen by Kathryn Cheah as the Senior Editor.

Essential revisions:

All three reviewers agreed that the ability of Atoh1 and Ikzf2 to drive supporting cells towards outer hair cell fates are interesting and represent an important milestone in hearing restoration research. We felt the number of samples used in the RNA-seq results were small and could confound the interpretations of the results. Considering additional source material is difficult and time consuming to obtain, advice from experts on how to gain more confidence in the analyses conducted was solicited. Please refer to Reviewer#3's comments for specifics. It is important that the suggestions be followed closely.

*Reviewer #1 (Recommendations for the authors):*

– IHCs overexpressing Atoh1 and Ikzf2 express Prestin, do they express other OHC properties? Do they lose IHC properties?

– Figure 2, make the shape of arrowhead more asymmetric in F-F" so it will be easier to tell what the authors are pointing at for F and F'.

– Figure 4G would benefit from a z-axis view of OHC-like cell in HC and SC levels.

*Reviewer #2 (Recommendations for the authors):*

The authors note almost as an aside that inducing Ikzf2 alone in SCs causes a loss of the endogenous OHCs. This non-autonomous effect is not explored or discussed. It makes one wonder why co-expression of Atoh1 with Ikzf2 in this experimental series was inefficient in inducing OHC-like production, since that worked when OHCs were deliberately killed. Is there something different about the different modes of cell killing? Or is something different about the timing of transcription factor expression and OHC expression? This issue should be addressed.

The confocal imaging data in this paper are beautiful and convincing. The characterization of the new mouse strains is extensive, but the authors should be sure to indicate a public repository for their mice, so that others can make use of them.

The P2A-DTR knocked into the Prestin stop codon strain represents a significant advance over the Pou4f3-DTR strain, which kills all HCs, and is often reported to have lethal off-target effects. The description of the characterization of this strain could be improved by adding information whether there are similar off-target effects.

The fact that the OHC-like cells are more like nascent hair cells could be due to the inability to down regulate Atoh1, which is only a transient factor in normal HC development. The authors need to address this possibility.

The scRNA-seq results did not offer a clear path forward. The authors suggest Emx2 as a possible factor to test, but the rationale is not well developed.

Line 120: Please change "and did not display any apparent phenotypes" to something like "and did not display any abnormalities" or "and did not display any abnormal phenotypes". Every genetic condition has a phenotype, which can be normal or abnormal.

Line 126: For clarity "wild-type OHCs" should be changed to "endogenous OHCs" or "pre-existing OHCs".

Line 168: Technically at this point it is not been demonstrated that the system is suitable for expressing Ikzf2 in adult SCs, it would be best to say that the system is suitable for expressing Ikzf2 in Cre-expressing cells.

Line 176: Although expression of tdTom indicates that the first part of the bicistonic transcript was translated, it does not guarantee that the first protein (Atoh1 in this case) is stable. Please provide information on HA expression or if it was already demonstrated in the previous work, just edit this description to clarify that you are sure that Atoh1 is being expressed.

Lines 358-359: Why were the wild type HCs and SCs picked at different ages (P30 vs. P60, respectively)?

Lines 988-989 (Figure 3 legend). This title makes it sound like we are looking at two different means of killing cells, one that is genetic and one that is a drug, when I think what is meant is that DT is the "pharmacologic" agent that triggers killing in the correct genetic background. Please edit to clarify.

Figure 1, Panel D: The legend for this panel does not indicate the type of error bars, nor the statistical test that produced the p-values. Later figures are also lacking the information on the statistical test. If this is the style of the journal, OK, but most readers would like to see the information.

Figure 2, Panel F: The variation should probably be described as not statistically significant, if that is the case.

Figure 4, Panels G: If all of the induced OHC-like cells in a single could be indicated, this would be helpful. Otherwise, it could appear that there is only one induced OHC-like cell in this view.

Figure 4, Panel H: The Y-axis title would be clearer if it were "% SCs transformed to OHC-like".

Figure 6, It is not clear why the color scheme for panel F and G are not the same.

*Reviewer #3 (Recommendations for the authors):*

1. The authors should perform unsupervised graph-based clustering of the data shown in Figure 6D to see if the cells that are outlined cluster with the OHCs. The analysis may have been performed but needs to be shown in the main figures.

2. Although the three cell types may not segregate cleanly, they do seem to spread out in interesting ways in Figure 6D, with more nascent HCs and "failed " SCs closer to the SC cluster and more OHC-like cells closer to the OHC cluster. This is encouraging and supports their main point. More interestingly, this spread is suggestive of a developmental trajectory, with intermediate cell states. With this in mind, they should perform a trajectory analysis, using a tool such as Monocle or Cell Trails, agnostic to their own assessment of cell type. It is also important to show that this spread is not due to technical heterogeneity, such as the number of genes detected, sequencing depth, or the proportion of mitochondrial genes. The authors should overlay this information on the UMAP plot in Figure 6D to demonstrate whether the spread is caused by technical artifacts.

3. The comparison to published data is informative and seems to have been performed appropriately using new tools in Seurat. It is strange, though, that the "semi-converted" cells overlap with only one of the P1 OHC clusters. The authors should add some more information about what distinguishes those clusters and how they interpret the fact that their cells overlap with one and not the other. These data would also be shown more appropriately as trajectories, again as determined using tools such as Monocle, Cell Trails, or Waterfall. Along these lines, the authors should include all of the data from the Kelley lab, not just the subsets they selected in Figure 6 supplemental Figure 4. The whole point is to place the converting cells within the context of normally developing hair cells, so it is best to include cells at multiple stages in this comparison.

4. The small samples sizes limit what one can conclude about differential gene expression. The problem is that there can be a very high false discovery rate when there are so few cells in each population. It would be better to lower the cut-off to an FDR-adjusted p-value of 0.001 or even lower. This would provide a more accurate view of the nature of differential gene expression for all of the comparisons. It is likely that the numbers of differentially expressed genes (thousands when comparing SCs and HCs and hundreds when comparing semi-converted cells to WT cells) will decrease with a more rigorous cut-off and thus provide a more accurate view of gene expression differences.

5. Along the same lines, rather than repeating the analysis in Figure 6E with other cell types, the authors should simply document the expression of known IHC or vestibular HC enriched genes. Presumably, there isn't a big enough cohort of non-OHC genes (either the number of genes or the level of expression) to drive the converting cells out of the OHC cluster. However, this does not mean that highly relevant genes associated with other hair cell types are not expressed at all. A reasonable list of genes can be compiled from the literature and then shown in a heat map for the three cell types. This is particularly important given that Atoh1 acts across hair cell types and there is a substantial Prestin-negative population. It is important to know if there is some heterogeneity here, in terms of the nature of HC identity taken on.

6. The authors need to soften their discussion of the GO analysis. Many of the terms they mention in the text are not actually statistically significant (i.e. p<0.05) in Figure 6 (Figure supp 1) and Figure 6 (supp 2). Only terms that are significantly enriched should be discussed.

7. I am surprised that only ABR results are reported, when DPOAEs offer a specific measure of OHC function. However, this would not significantly alter conclusions, would just be a nice confirmation.

8. Line 62: "transition" is not the right word to use here. Perhaps the authors mean "transmission"?

9. Line 1072: "compliment" should be "complement".

10. Figure 6- supplement 4: Please explain what the dots labeled 1 and 2 indicate. I found this figure hard to understand. There is also a typo in "apical". There should also be a clearer explanation of what the "pseudotime" scale is (arbitrary units??).

11. Line 537: "or to repress the IHC fate": this doesn't need to be an either/or statement. Many transcription factors both induce one fate and repress another.

12. Line 642: There is an extra "were" in this sentence.

13. Line 643: "whole-amount" should be "whole-mount".

[Editors' note: further revisions were suggested prior to acceptance, as described below.]

Thank you for resubmitting your work entitled "Dual expression of Atoh1 and Ikzf2 promotes transformation of adult cochlear supporting cells into outer hair cells" for further consideration by *eLife*. Your revised article has been evaluated by Kathryn Cheah (Senior Editor) and a Reviewing Editor.

All three reviewers thought you have made a good-faith effort in incorporating all the suggestions. However, there are still some outstanding issues from all three reviewers, and the Reviewing Editor has drafted this to help you prepare a revised submission.

This manuscript demonstrated the effectiveness of combined activation of Atoh1 and Ikzf2 in converting adult supporting cells to outer hair cell (OHC)-like cells in a mouse model, in which the OHCs have been selectively ablated with diphtheria toxin. The authors showed that while the number of regenerated hair cells was low and there was no functional recovery based on ABR, these OHC-like cells do express prestin and exhibit a genetic profile that resemble nascent hair cells. This paper will be of great interest to researchers interested in hearing restoration as well as regenerative biology.

Essential revisions:*Reviewer #1:*

Figure 4 still needs improvement. I thought a z-axis view will help readers to see where the OHC-like cells are located, at the normal HC level as depicted by Figure 4B or they remained at the SC level or both. The added insets of Z axis did not clarify this point. If the OHC-like cells were located at different positions, then better illustrations and quantification are needed.

The diagram in 4B could also be modified to indicate the OHC-like cells came from td-labeled SCs, using hatched black and red colors, for example.

*Reviewer #3:*

Overall, the authors have done a decent job of incorporating our suggested changes, but the changes do not persuade me that their conversion is as complete as they make it seem. I remain puzzled by the scRNA-seq results, which to me indicate a far from complete conversion that undermines the suggestion that they have even been able to make P1-ish OHCs from adult SCs. For instance, the new hierarchical clustering data in Figure 6 show that 5 cell types form 2 clusters, which tells me that there isn't enough power in the study to make fine distinctions, likely because of the nature of gene expression changes involved. The new trajectories underscore this point, in that the cells that receive Atoh1/Ikzf2 follow their own trajectory and just barely converge with the mature OHCs, not with the P1-ish OHCs. I was glad to see some plots of known IHC genes and one vestibular gene, but really they just show a handful of genes. It would have been much better to show a heat map of ~50 enriched markers for each. I would also be curious to see if the OHC-like cells are like P1 vestibular cells, with the OHC nature driven mostly by expression of Prestin. Also, the new data for oncomodulin expression and synaptic ribbon number are not germane to the argument, as they are shown only for overexpression of Ikzf2 alone in IHC; the heat maps in Figure 6- supplement 6 in fact show that oncomodulin is just barely induced in a handful of OHC-like cells. The discussion contradicts this data and states that the OHC-like cells do express oncomodulin (p. 28, line 19), so this needs to be clarified.

To be clear, I do not think that it is reasonable to expect complete conversion of mature SCs into functional OHCs upon sustained overexpression of Atoh1 and Ikzf2. In the regeneration field, one can learn just as much from incomplete conversions and in fact, that is an important goal since it tells us how far certain combinations can take us while uncovering new roadblocks that we should tackle in the future. The authors made a good attempt to discuss this at the end of the discussion. My request to the authors is to be more restrained in how they present these data. I am not yet convinced that the OHC-like cells are basically like P1 OHCs so much as hybrid HC/SCs that have been able to turn on a few developmentally regulated HC genes and enough prestin to make them cluster with OHCs. I feel that the extent of the conversion needs to be discussed more explicitly, maybe just with a quick summary paragraph outlining how many OHC genes come up, how many SC genes go down, what key genes are missing, plus the extent of similarity to P1 OHCs (how many genes are differentially expressed between P1 OHC and their OHC-like cells and what are those genes?). The authors make some interesting points, such as lack of expression of functionally relevant genes, but still don't provide any meta-analysis that could help others to think about what might be limiting the extent of conversion. I would like to see a more concrete discussion of what the results mean, aside from what we already knew (Atoh1 is good at turning on HC genes and Ikzf2 is pretty good at turning on Prestin).

Finally, all three reviewers thought the manuscript could use some in-depth editing from a trusted colleague who understands the science as well as a master of the English language.

[Editors' note: further revisions were suggested prior to acceptance, as described below.]

Thank you for resubmitting your work entitled "Dual expression of Atoh1 and Ikzf2 promotes transformation of adult cochlear supporting cells into outer hair cells" for further consideration by *eLife*. Your revised article has been evaluated by Kathryn Cheah (Senior Editor) and a Reviewing Editor.

The manuscript has been improved but there are some remaining issues that need to be addressed, as outlined below:

Essential revisions:

1) The RNA seq data are acceptable. However, some sentences need clarification.

Line 569, We concluded that OHC-like cells were, based on their characteristic gene profiles, unlike neonatal IHCs or utricle HCs, instead ~50% similar with neonatal OHCs.

This sentence is awkward, and it is not clear whether the approximately 50% meant 50% in similarity between OHC-like cells and neonatal OHCs based on the selected 53 OHC genes analyzed or 50% of the OHC-like cells were more like the neonatal OHCs. Either clarify or simplify. For example: Taken together all the gene profiling results, we concluded that OHC-like cells resemble neonatal OHCs and not neonatal IHCs or utricle HCs.

Similar clarification is needed for the sentence on line 726.

2) The positions of OHC-like cells within the epithelium still need further clarification. Despite the nice video and the added supplemental figure, it remains difficult to distinguish the location of all the OHC-like cells shown in Figure 4G. A simple fix would be to use the blue and gold color arrows in Figure 4G to point out which are the HCs located in the HC layer versus SC layer. The supplemental figure can then be used to illustrate the two different locations of HCs in more detail.

Line 340, In addition, ~82.6% of OHC-like cells (i.e. arrows in Figure 4G') migrated up (toward the HC layer), and the rest remaining in the bottom (SC layer).

Is it necessary to instil the phrase of "migrated up" considering the cellular organization of the epithelium appears to be somewhat disrupted? "Migrated-up" suggests some active process on the part of the OHC-like cells. Consider replacing the sentence with: "In addition, ~82.6% of OHC-like cells were located in the top HC layer (blue arrows in Figure 4G', supple figure), whereas the rest remained in the bottom SC layer (gold arrows in Figure 4G", Supl figure ).

---

## [Author Response]

Essential revisions:All three reviewers agreed that the ability of Atoh1 and Ikzf2 to drive supporting cells towards outer hair cell fates are interesting and represent an important milestone in hearing restoration research. We felt the number of samples used in the RNA-seq results were small and could confound the interpretations of the results. Considering additional source material is difficult and time consuming to obtain, advice from experts on how to gain more confidence in the analyses conducted was solicited. Please refer to Reviewer#3's comments for specifics. It is important that the suggestions be followed closely.Reviewer #1 (Recommendations for the authors):– IHCs overexpressing Atoh1 and Ikzf2 express Prestin, do they express other OHC properties? Do they lose IHC properties?

We additionally show that IHCs overexpressing Ikzf2 express another OHC specific marker, Oncomodulin (Ocm). Furthermore, to demonstrate what IHC properties are lost, co-staining of Slc7a14 (another IHC marker) and Ctbp2 (a marker of synaptic ribbons) shows that, relative to control IHCs, IHCs overexpressing Ikzf2 lose Slc7a14 expression, have less Ctbp2+ puncta and smaller cell nuclei. Please refer to lines 4-18 on page 8 and the added Figure 1—figure supplement 3.

– Figure 2, make the shape of arrowhead more asymmetric in F-F" so it will be easier to tell what the authors are pointing at for F and F'.

Thanks for this suggestion. We agree that asymmetric labelling is easier to visualize the targeted cell. Please see the updated Figure 2.

– Figure 4G would benefit from a z-axis view of OHC-like cell in HC and SC levels.

We agree that adding a z-axis view can provide more information. Please see the inset images in Figure 4G-G’’’.

Reviewer #2 (Recommendations for the authors):The authors note almost as an aside that inducing Ikzf2 alone in SCs causes a loss of the endogenous OHCs. This non-autonomous effect is not explored or discussed. It makes one wonder why co-expression of Atoh1 with Ikzf2 in this experimental series was inefficient in inducing OHC-like production, since that worked when OHCs were deliberately killed. Is there something different about the different modes of cell killing? Or is something different about the timing of transcription factor expression and OHC expression? This issue should be addressed.

Thanks for pointing it out. We added a section where we provided our speculations in the discussion part. Please refer to lines 21-22 on page 26, lines 1-22 on page 27, and lines 1-3 on page 28.

The confocal imaging data in this paper are beautiful and convincing. The characterization of the new mouse strains is extensive, but the authors should be sure to indicate a public repository for their mice, so that others can make use of them.

Thanks for the encouraging comments. We have contacted the Jackson Lab in which the *Prestin-DTR* and *Rosa26-LSL-Ikzf2* strains would be deposited. The JAX Stock numbers are: 036271 and 036272, respectively. This information is included in the “key resource table” of the revised manuscript. Please refer to page 32.

The P2A-DTR knocked into the Prestin stop codon strain represents a significant advance over the Pou4f3-DTR strain, which kills all HCs, and is often reported to have lethal off-target effects. The description of the characterization of this strain could be improved by adding information whether there are similar off-target effects.

Besides OHC death and deafness, we do not observe any other apparent defects in *Prestin-DTR/+* mice (it is now changed to *Slc26a5^DTR/+^*, as requested by the *eLife* office editor)*.* We have added a sentence describing the minimal off-target effects in *Prestin-DTR/+* mice. Please refer to line 1-3 on page 13.

The fact that the OHC-like cells are more like nascent hair cells could be due to the inability to down regulate Atoh1, which is only a transient factor in normal HC development. The authors need to address this possibility.

We agree that permanent expression of Atoh1 is a possible barrier preventing OHC-like cells from maturing. We realize that this issue should be resolved in future, and indeed our lab are in the process of optimizing a genetic approach by which conditional, but transient Atoh1 can be successfully induced in adult SCs. Please refer to lines 21-22 on page 29, and lines 1-5 on page 30.

The scRNA-seq results did not offer a clear path forward. The authors suggest Emx2 as a possible factor to test, but the rationale is not well developed.

We have re-organized the text to explain why we are interested in genes involved in hair bundle organization. Please refer to lines 1-5 on page 31.

Line 120: Please change "and did not display any apparent phenotypes" to something like "and did not display any abnormalities" or "and did not display any abnormal phenotypes". Every genetic condition has a phenotype, which can be normal or abnormal.

We accept the suggestion and choose “and did not display any abnormalities”. Please refer to line 7 on page 6.

Line 126: For clarity "wild-type OHCs" should be changed to "endogenous OHCs" or "pre-existing OHCs".

We have changed “wild-type OHCs" to "endogenous OHCs". Please refer to line 13 on page 6.

Line 168: Technically at this point it is not been demonstrated that the system is suitable for expressing Ikzf2 in adult SCs, it would be best to say that the system is suitable for expressing Ikzf2 in Cre-expressing cells.

We agree with this suggestion and please refer to line 2 on page 9.

Line 176: Although expression of tdTom indicates that the first part of the bicistonic transcript was translated, it does not guarantee that the first protein (Atoh1 in this case) is stable. Please provide information on HA expression or if it was already demonstrated in the previous work, just edit this description to clarify that you are sure that Atoh1 is being expressed.

The efficient Atoh1 expression in *CAG-LSL-Atoh1+* has been demonstrated in our previous work (Liu et al., JNS, 2012). We have re-written this sentence and make it clear. Please refer to lines 11-14 on page 9.

Lines 358-359: Why were the wild type HCs and SCs picked at different ages (P30 vs. P60, respectively)?

Thanks for pointing it out. We decide to pick wild type OHCs at P30, because we roughly consider the intrinsic age (counting from reprogramming starts) of the OHC-like cells as P30. In other words, although cochlear samples are analyzed at P60, new OHC-like cells have experienced reprogramming for only 30 days. Therefore, we speculate it is more appropriate to compare transcriptomic profiles between OHC-like cells and P30, rather than P60, wild type OHCs.

Nevertheless, as cochlear development is complete by P30, the transcriptomic profiles of cochlear cells should be stable or minimally changed between P30 and P60. This notion is supported by a recent study of RNA-Seq of adult cochlear SCs at P60 and P120. Please refer to lines 14-17 on page 28.

Lines 988-989 (Figure 3 legend). This title makes it sound like we are looking at two different means of killing cells, one that is genetic and one that is a drug, when I think what is meant is that DT is the "pharmacologic" agent that triggers killing in the correct genetic background. Please edit to clarify.

Thanks for pointing it out. We have updated the title of Figure 3 legend as “Damaging adult OHCs specifically by DT treatment”. Please refer to line 1 on page 50.

Figure 1, Panel D: The legend for this panel does not indicate the type of error bars, nor the statistical test that produced the p-values. Later figures are also lacking the information on the statistical test. If this is the style of the journal, OK, but most readers would like to see the information.

Statistic and error bars information were previously described in method section only. We have added them in Figure legends for better understanding. Please refer to lines 11-12 on page 48, lines 15-16 on page 49, lines 12-13 on page 50, lines 14-15 on page 51, and lines 10-11 on page 52.

Figure 2, Panel F: The variation should probably be described as not statistically significant, if that is the case.

Yes, there was no statistic difference, and we have marked it in the updated Figure 2G.

Figure 4, Panels G: If all of the induced OHC-like cells in a single could be indicated, this would be helpful. Otherwise, it could appear that there is only one induced OHC-like cell in this view.

We have accepted this suggestion. To simplify the Figure 4, all 6 OHC-like cells were labelled with blue and yellow arrows in Figure 4G’’ only, except that the OHC-like cell labelled by blue arrows were highlighted in Figure 4G-G’’’, because it was also visualized in YZ axis (inset), as suggested by reviewer 1.

Figure 4, Panel H: The Y-axis title would be clearer if it were "% SCs transformed to OHC-like".

We have accepted this suggestion.

Figure 6, It is not clear why the color scheme for panel F and G are not the same.

In the updated Figure 6, we mixed 8 different cell populations for unsupervised cell cluster. Panel F show that 5 main clusters are formed, meaning that some of the 8 different cell populations are similar and are classified in the same cluster. Basically, panel G is same as Panel F, except that each of the 8 cell populations are assigned with different colors in the 5 main clusters.

Reviewer #3 (Recommendations for the authors):1. The authors should perform unsupervised graph-based clustering of the data shown in Figure 6D to see if the cells that are outlined cluster with the OHCs. The analysis may have been performed but needs to be shown in the main figures.

Thanks for pointing it out. The unsupervised graph-based clustering is now included in the updated Figure 6D.

2. Although the three cell types may not segregate cleanly, they do seem to spread out in interesting ways in Figure 6D, with more nascent HCs and "failed " SCs closer to the SC cluster and more OHC-like cells closer to the OHC cluster. This is encouraging and supports their main point. More interestingly, this spread is suggestive of a developmental trajectory, with intermediate cell states. With this in mind, they should perform a trajectory analysis, using a tool such as Monocle or Cell Trails, agnostic to their own assessment of cell type. It is also important to show that this spread is not due to technical heterogeneity, such as the number of genes detected, sequencing depth, or the proportion of mitochondrial genes. The authors should overlay this information on the UMAP plot in Figure 6D to demonstrate whether the spread is caused by technical artifacts.

Thanks for this suggestion. We have performed trajectory analysis via Monocle. Indeed, the OHC-like cells/nascent HCs are also closer to wild type OHC cluster along the developmental trajectory line. Please refer to the added Figure 6—figure supplement 3. Furthermore, we also include the raw reads genome mapping rate, gene detection numbers (TPM>1 and TPM>10) in the Supplementary file 8. Basically, we confirm the technical heterogeneity among our Smartseq library is minimal.

3. The comparison to published data is informative and seems to have been performed appropriately using new tools in Seurat. It is strange, though, that the "semi-converted" cells overlap with only one of the P1 OHC clusters. The authors should add some more information about what distinguishes those clusters and how they interpret the fact that their cells overlap with one and not the other. These data would also be shown more appropriately as trajectories, again as determined using tools such as Monocle, Cell Trails, or Waterfall. Along these lines, the authors should include all of the data from the Kelley lab, not just the subsets they selected in Figure 6 supplemental Figure 4. The whole point is to place the converting cells within the context of normally developing hair cells, so it is best to include cells at multiple stages in this comparison.

We agree with and accept this good suggestion. In the revised manuscript, we pool all OHCs covered by the data from the Kelley lab, with our manually picked cells. Trajectory analysis is performed via Monocle. Consistent with previous analysis, OHC-like cells and nascent cells primarily overlap with only one of the P1 OHC clusters which should be more differentiated OHCs (middle and basal turns) because they are closer to P7 and P30_WT OHCs. Cluster 4 should be P1 OHCs at apical turns which are less differentiated and closer to E16_WT OHCs. Please refer to lines 20-21 on page 22, and lines 12-15 on page 23 and the updated Figure 6.

4. The small samples sizes limit what one can conclude about differential gene expression. The problem is that there can be a very high false discovery rate when there are so few cells in each population. It would be better to lower the cut-off to an FDR-adjusted p-value of 0.001 or even lower. This would provide a more accurate view of the nature of differential gene expression for all of the comparisons. It is likely that the numbers of differentially expressed genes (thousands when comparing SCs and HCs and hundreds when comparing semi-converted cells to WT cells) will decrease with a more rigorous cut-off and thus provide a more accurate view of gene expression differences.

We have switched to use FDR-adjusted p-value of 0.001, as clearly described in method section (lines 5-7 on page 38). Accordingly, the numbers of different genes are decreased, GO analysis and the corresponding figures and excel files containing list of all differently expressed genes are updated.

5. Along the same lines, rather than repeating the analysis in Figure 6E with other cell types, the authors should simply document the expression of known IHC or vestibular HC enriched genes. Presumably, there isn't a big enough cohort of non-OHC genes (either the number of genes or the level of expression) to drive the converting cells out of the OHC cluster. However, this does not mean that highly relevant genes associated with other hair cell types are not expressed at all. A reasonable list of genes can be compiled from the literature and then shown in a heat map for the three cell types. This is particularly important given that Atoh1 acts across hair cell types and there is a substantial Prestin-negative population. It is important to know if there is some heterogeneity here, in terms of the nature of HC identity taken on.

Thanks for this suggestion. We choose several reported OHC, IHC and vestibular HC specific genes and plot their TPM values in all the 11 OHC-like cells. In agreement with our UMAP analysis, OHC specific genes Insm1, Insm2, and St8sia3 are highly expressed in OHC-like cells, however, IHC specific genes Fgf8, Otof, Slc17a8, Slc7a14, Tbx2 and vestibular HC gene Zpld1 are minimally or not expressed in OHC-like cells. Again, it confirms that OHC-like cells do resemble neonatal endogenous OHCs. Please refer to lines 5-15 on page 21 and Figure 6—figure supplement 2C-E.

6. The authors need to soften their discussion of the GO analysis. Many of the terms they mention in the text are not actually statistically significant (i.e. p<0.05) in Figure 6 (Figure supp 1) and Figure 6 (supp 2). Only terms that are significantly enriched should be discussed.

We have updated our analysis standard in the revised manuscript and only discussed the GO analysis of genes that are statistically different (FDR-adjusted p-value of 0.001).

7. I am surprised that only ABR results are reported, when DPOAEs offer a specific measure of OHC function. However, this would not significantly alter conclusions, would just be a nice confirmation.

We agree that DPOAE are more specific for OHC function measurement. We did not include it only because we are not confident yet of our current DPOAE assays in the lab. To validate our DPOAE assay, we will practice more by using mutant mice with OHC specific deficiency (e.g. Prestin -/- mice). Thanks for pointing it out, and we certainly will incorporate DPOAE measurement in our future OHC regeneration studies.

8. Line 62: "transition" is not the right word to use here. Perhaps the authors mean "transmission"?

Thanks for pointing it out. We have changed "transition" to "transmission". Please refer to line 13 on page 3.

9. Line 1072: "compliment" should be "complement".

Sorry for the typo and "compliment" is changed to "complement". Please refer to line 5 on page 54.

10. Figure 6- supplement 4: Please explain what the dots labeled 1 and 2 indicate. I found this figure hard to understand. There is also a typo in "apical". There should also be a clearer explanation of what the "pseudotime" scale is (arbitrary units??).

This figure is updated because all OHCs (from Kelley lab paper) are included and now it is Figure 6- supplement 5. In addition, the previous dots labeled 1 and 2 indicates the developmental bifurcation, but we have removed it in the updated Figure because the bifurcation is apparent. The typo is also corrected. Last, the pseudotime scale is an arbitrary unit, and it is simulated according to the developmental order (or age/differentiation status) of each cell. Please refer to lines 9-11 on page 39.

11. Line 537: "or to repress the IHC fate": this doesn't need to be an either/or statement. Many transcription factors both induce one fate and repress another.

We have changed the “or” to “and”. Please refer to line 10 on page 26.

12. Line 642: There is an extra "were" in this sentence.

Thanks for your careful reading. We have deleted the second “were”.

13. Line 643: "whole-amount" should be "whole-mount".

Thanks, and "whole-amount" is changed to "whole-mount".

[Editors' note: further revisions were suggested prior to acceptance, as described below.]

Essential revisions:Reviewer #1:Figure 4 still needs improvement. I thought a z-axis view will help readers to see where the OHC-like cells are located, at the normal HC level as depicted by Figure 4B or they remained at the SC level or both. The added insets of Z axis did not clarify this point. If the OHC-like cells were located at different positions, then better illustrations and quantification are needed.

We provided a new Figure 4—figure supplement 1 where confocal XY, YZ, and XZ angles were presented. Furthermore, to better visualize those OHC-like cells at various angles, the same confocal raw data (Figure 4 and Figure 4—figure supplement 1) were subjected to Imaris software and a rich media file was produced. Please refer to lines 12-13 on page 15.

The diagram in 4B could also be modified to indicate the OHC-like cells came from td-labeled SCs, using hatched black and red colors, for example.

Thanks, and we have accepted the suggestion and modified the diagram in Figure 4B. The new OHC-like cells were currently labelled with hatched back and red colors.

Reviewer #3:Overall, the authors have done a decent job of incorporating our suggested changes, but the changes do not persuade me that their conversion is as complete as they make it seem. I remain puzzled by the scRNA-seq results, which to me indicate a far from complete conversion that undermines the suggestion that they have even been able to make P1-ish OHCs from adult SCs. For instance, the new hierarchical clustering data in Figure 6 show that 5 cell types form 2 clusters, which tells me that there isn't enough power in the study to make fine distinctions, likely because of the nature of gene expression changes involved. The new trajectories underscore this point, in that the cells that receive Atoh1/Ikzf2 follow their own trajectory and just barely converge with the mature OHCs, not with the P1-ish OHCs. I was glad to see some plots of known IHC genes and one vestibular gene, but really, they just show a handful of genes. It would have been much better to show a heat map of ~50 enriched markers for each. I would also be curious to see if the OHC-like cells are like P1 vestibular cells, with the OHC nature driven mostly by expression of Prestin.

Thanks for the encouraging comments. First, our previous analysis criteria/resolution might underestimate the power of the transcriptomic difference among our manual picked cells. Indeed, the 5 cell types in Figure 6E was able to be divided into 4 main clusters (updated Figure 6D), but the key conclusion was the same.

Second, we agreed with the comments about how to describe the OHC-like cells. In the 2^nd^ revised manuscript, we clarified that we did not intend to conclude that OHC-like cells were fully equivalent to P1-OHCs, instead their similarity was ~50%.

Third, following the suggestions, we provided a heatmap where 49 neonatal cochlear IHC and 26 neonatal utricle HC specific genes were selected. The number of IHC specific genes was close to 50, but number of utricle HC specific genes was roughly half. Those genes were selected by overlapping cell type specific genes reported in three different studies (Burns, et al., *Nat Commun*, 2015, PMID: 26469390; Wiwatpanit, et al., *Nature*, 2018, PMID: 30305733; Kolla, et al., *Nat Commun*, 2020, PMID: 32404924). Their different sequencing methods or other bench factors would cause variations. In other words, if one gene is only cell type specific in one or two reports, it was not regarded as IHC or utricle HC specific genes. This standard led to reduced numbers of cell type specific genes, but in our mind would make the analysis more accurate. Please refer to result section “Neonatal specific OHC genes are highly expressed in OHC-like cells” starting from line 14 on page 24.

Notably, we remind the readers that we also feel it is challengeable and has limitations to compare data from different studies, especially the data are generated by different sequencing methods, as the similar comment is alerted (Kolla, et al., *Nat Commun*, 2020, PMID: 32404924). In other words, the field urgently needs a complete transcriptomic profile produced by smartseq or other full-length cDNA sequencing approaches that covers distinct inner ear HC subtypes at different developmental ages, because adult inner ear HCs are hard to perform via 10x single cell sequencing platform. Given this dataset is available, it is easier and more precise to define what the regenerated HCs are in future studies. Please see text on lines 18-22 on page 32 and line 1 on page 33.

Also, the new data for oncomodulin expression and synaptic ribbon number are not germane to the argument, as they are shown only for overexpression of Ikzf2 alone in IHC; the heat maps in Figure 6- supplement 6 in fact show that oncomodulin is just barely induced in a handful of OHC-like cells. The discussion contradicts this data and states that the OHC-like cells do express oncomodulin (p. 28, line 19), so this needs to be clarified.

To clarify this issue, we double checked the oncomodulin (Ocm) expression level in the 11 OHC-like cells. It seemed low in previous heatmap primarily because the TPM value in the P30_WT OHC was high. In the current revised version, we included the log- transformed transcripts per million (TPM) value of *Ocm* in each OHC-like cell in the updated Figure 6—figure supplement 6B (heatmap) and the raw TPM value in Supplementary file 6.

Indeed, 7/11 new OHC-like cells expressed high level of Ocm. Consistently, Ocm was not included in the differently expressed genes between OHC-like cells and adult endogenous OHCs (updated Figure 6—figure supplement 7A). We modified Figure 6—figure supplement 7A (previously as Figure 6—figure supplement 6A) to better help readers to more clearly appreciate the differentially expressed SC-or OHC-genes between OHC-like cells and adult OHCs. Please also see text on lines 19-21 on page 31.

To be clear, I do not think that it is reasonable to expect complete conversion of mature SCs into functional OHCs upon sustained overexpression of Atoh1 and Ikzf2. In the regeneration field, one can learn just as much from incomplete conversions and in fact, that is an important goal since it tells us how far certain combinations can take us while uncovering new roadblocks that we should tackle in the future. The authors made a good attempt to discuss this at the end of the discussion. My request to the authors is to be more restrained in how they present these data. I am not yet convinced that the OHC-like cells are basically like P1 OHCs so much as hybrid HC/SCs that have been able to turn on a few developmentally regulated HC genes and enough prestin to make them cluster with OHCs. I feel that the extent of the conversion needs to be discussed more explicitly, maybe just with a quick summary paragraph outlining how many OHC genes come up, how many SC genes go down, what key genes are missing, plus the extent of similarity to P1 OHCs (how many genes are differentially expressed between P1 OHC and their OHC-like cells and what are those genes?).

Thanks and we accepted those suggestions. Thus, we added a section to explicitly discuss the extent of cell fate conversion, similarity between OHC-like cells and P1-OHCs as well as how to define the converted OHC-like cells. Please refer to the section of “degree of cell fate conversion of the OHC-like cells” in the discussion part starting at line 19 on page 30.

In addition, please also refer to the new Figure 7 where we briefly summarized the key information, which included how many OHC genes come up and how many SC genes go down in OHC-like cells, relative to adult SCs and OHCs, plus the ~50% similarity to P1 OHCs, and minimal similarity to neonatal IHCs and utricle HCs. The key missing genes were also summarized in summarized in Supplementary files 6 and 7.

The authors make some interesting points, such as lack of expression of functionally relevant genes, but still don't provide any meta-analysis that could help others to think about what might be limiting the extent of conversion. I would like to see a more concrete discussion of what the results mean, aside from what we already knew (Atoh1 is good at turning on HC genes and Ikzf2 is pretty good at turning on Prestin).

Besides what we have included previously, we have further discussed the barriers that preventing OHC-like cells from undergoing further differentiation. Two main points were added after meta-analysis:

1) *Bcl11b* was missing in OHC-like cells and Bcl11b is involved in stereocilia development in OHCs. We have speculated that *Bcl11b* might be one candidate gene to be combined with Atoh1 and Ikzf2 in future. Please refer to text on lines 4-8 on page 34.

2) *Fgfr3* and *Id1/2* families were expressed much higher in OHC-like cells than in adult endogenous OHC cells. They are involved in either repressing Atoh1 or maintaining SC fate, thus we speculated that further decreasing *Fgfr3* and *Id1/2* would further promote the conversion extent of OHC-like cells. Please refer to text on lines 3-8 on page 35.

Finally, all three reviewers thought the manuscript could use some in-depth editing from a trusted colleague who understands the science as well as a master of the English language.

I have asked my previous colleague, Ms.Virginia M. S. Rutten (a senior graduate in HHMI-Janelia Research Campus, Ashburn, VA, USA) for assistance in editing the text.

[Editors' note: further revisions were suggested prior to acceptance, as described below.]

The manuscript has been improved but there are some remaining issues that need to be addressed, as outlined below:Essential revisions:1) The RNA seq data are acceptable. However, some sentences need clarification.Line 569, We concluded that OHC-like cells were, based on their characteristic gene profiles, unlike neonatal IHCs or utricle HCs, instead ~50% similar with neonatal OHCs.This sentence is awkward, and it is not clear whether the approximately 50% meant 50% in similarity between OHC-like cells and neonatal OHCs based on the selected 53 OHC genes analyzed or 50% of the OHC-like cells were more like the neonatal OHCs. Either clarify or simplify. For example: Taken together all the gene profiling results, we concluded that OHC-like cells resemble neonatal OHCs and not neonatal IHCs or utricle HCs.Similar clarification is needed for the sentence on line 726.

Thanks for pointing this out. The 50% similarity means that, based on the selected 53 OHC genes analyzed, the overall similarity between all the 11 OHC-like cells and neonatal OHCs is ~50%. Please refer to the updated sentences on lines 15-17, page 25.

We also accepted the suggested simplified sentence: “*Taken together all the gene profiling results, we concluded that OHC-like cells resemble neonatal OHCs and not neonatal IHCs or utricle HCs”.* Please refer to lines 9-11, page 26 and lines 7-8, page 33.

2) The positions of OHC-like cells within the epithelium still need further clarification. Despite the nice video and the added supplemental figure, it remains difficult to distinguish the location of all the OHC-like cells shown in Figure 4G. A simple fix would be to use the blue and gold color arrows in Figure 4G to point out which are the HCs located in the HC layer versus SC layer. The supplemental figure can then be used to illustrate the two different locations of HCs in more detail.

Thanks for this suggestion and we accepted it. Among all the 6 OHC-like cells in the updated Figure 4G-G’’’, 5 of them (blue arrows) are located in the HC layer and one (orange arrow) is in SC layer. The OHC-like cell (blue arrow in Figure 4G’’’) is presented in the orthogonal views (Figure 4—figure supplement 1A-A’’’).

We chose another OHC-like cell that is located in a further deeper layer (SC layer) than the one (orange arrow in Figure 4G’’), and its orthogonal views are presented in Figure 4—figure supplement 1B-B’’’. Comparison between Figure 4—figure supplement 1A-A’’’ and Figure 4—figure supplement 1B-B’’’, in our mind, currently clearly illustrates the different locations of the OHC-like cells.

Line 340, In addition, ~82.6% of OHC-like cells (i.e. arrows in Figure 4G') migrated up (toward the HC layer), and the rest remaining in the bottom (SC layer).Is it necessary to instill the phrase of "migrated up" considering the cellular organization of the epithelium appears to be somewhat disrupted? "Migrated-up" suggests some active process on the part of the OHC-like cells. Consider replacing the sentence with: "In addition, ~82.6% of OHC-like cells were located in the top HC layer (blue arrows in Figure 4G', supple figure), whereas the rest remained in the bottom SC layer (gold arrows in Figure 4G", Supl figure ).

We accepted the suggestion. Please refer to the suggested sentence in lines 20-22, page 15 and line 1, page 16.